# Self-Improving Transformers Overcome Easy-to-Hard and Length Generalization Challenges

**Nayoung Lee** [*1]  **Ziyang Cai** [*1]  **Avi Schwarzschild** [2]  **Kangwook Lee** [1]  **Dimitris Papailiopoulos** [1 3]

## Abstract

Large language models often struggle with length generalization and solving complex problem instances beyond their training distribution. We present a self-improvement approach where models iteratively generate and learn from their own solutions, progressively tackling harder problems while maintaining a standard transformer architecture. Across diverse tasks including arithmetic, string manipulation, and maze solving, our method enables models to solve problems far beyond their initial training distribution—for instance, generalizing from 10-digit to 100-digit addition without apparent saturation. We observe that filtering for correct self-generated examples leads to exponential improvements in out-of-distribution performance across training rounds. Additionally, starting from pretrained models significantly accelerates this self-improvement process for several tasks. Our results demonstrate how controlled weak-to-strong curricula can systematically expand model capabilities while preserving architectural simplicity.

## 1. Introduction

Despite the remarkable success of transformer-based language models (Vaswani et al., 2017) across a wide range of tasks, these models exhibit significant limitations in *length generalization*—the ability to extrapolate to longer sequences than those seen during training. Even in simple algorithmic tasks such as arithmetic, standard transformer models trained with autoregressive objectives struggle to generalize to longer problem instances (Anil et al., 2022).

To address this, prior work has explored various modifica-

tions, including changes to positional embeddings (Ruoss et al., 2023; McLeish et al., 2024; Kazemnejad et al., 2024; Li et al., 2023; Sabbaghi et al., 2024; Cho et al., 2024; Zhou et al., 2024), architectural modifications (Fan et al., 2024; Duan et al., 2023), and data format changes such as index hinting (Zhou et al., 2023; 2024). While effective in controlled setups, these approaches are often incompatible with large language models (LLMs) in practice, as they introduce task-specific modifications that are difficult to scale across diverse applications.

Instead of architectural modifications, we exploit transformers' tendency to exhibit some "transcendence" (Zhang et al., 2024) beyond their training distribution - models trained on simple task instances can sometimes solve slightly harder ones. Specifically, models trained on simple instances of a task can sometimes generate correct outputs for slightly harder instances. We leverage this property by applying a **self-improvement** framework, where the model iteratively generates its own training data and progressively learns from harder examples.

Self-improvement has been widely studied in various contexts (Singh et al., 2023; Gulcehre et al., 2023; Liang et al., 2024), typically in settings where external verifiers, weak supervision, or filtering mechanisms are used to ensure data quality. We demonstrate that extreme length generalization is possible under this framework, even *without architectural modifications*. For tasks like reverse addition and string copying, self-improvement succeeds with no explicit data filtering. However, for harder problems such as multiplication and shortest path finding in mazes, naive self-improvement fails due to error accumulation. We show that simple data filtering techniques—such as length filtering and majority voting—suffice to maintain data quality and enable self-improvement to extend far beyond the initial training distribution.

Our findings suggest that self-improvement is not limited to length generalization but also enables *easy-to-hard generalization*, where a model trained on simpler tasks successfully learns harder tasks without additional supervision. Notably, our approach does not introduce a new self-improvement framework but instead demonstrates its effectiveness across diverse algorithmic tasks.

---

[*]Equal contribution  [1]University of Wisconsin-Madison [2]Carnegie Mellon University [3]Microsoft. Correspondence to: Nayoung Lee <nayoung.lee@wisc.edu>, Ziyang Cai <zcai75@wisc.edu>.

*Proceedings of the 42nd International Conference on Machine Learning*, Vancouver, Canada. PMLR 267, 2025. Copyright 2025 by the author(s).

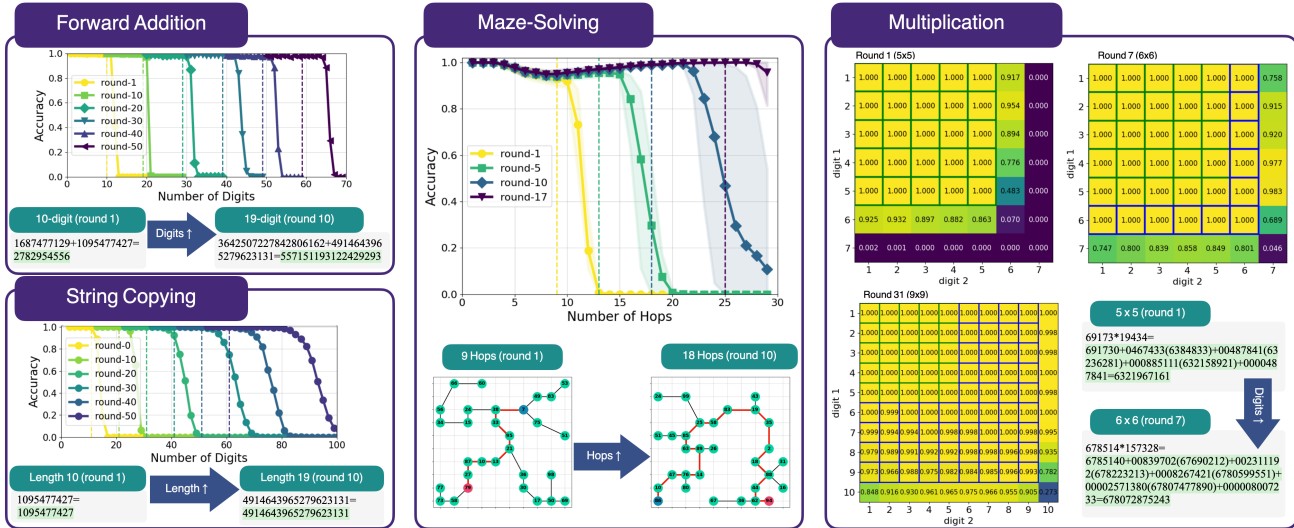

*Figure 1.* Overview of self-improvement results. Models trained with self-improvement can tackle increasingly complex tasks that extend far beyond their initial training distributions, achieving significant generalization **without any additional supervision**.

Furthermore, we investigate the dynamics of self-improvement and show that: (1) controlling the weak-to-strong curriculum is crucial, as models require a structured difficulty schedule to avoid catastrophic failure, (2) self-improvement accelerates over time, as models increasingly benefit from harder examples, leading to exponential extrapolation, and (3) pretrained models are better at self-improvement, generalizing further and faster than models trained from scratch. These results position self-improvement as a scalable solution for length generalization and beyond. Our contributions can be summarized as:

1. We apply an iterative self-training framework to train transformers on the arithmetic, maze and string manipulation tasks, and successfully tackle **easy-to-hard generalization** to extreme out-of-distribution test data.
2. We motivate the importance of a carefully crafted self-improvement schedule and **label filtering** based on length and majority voting, which are central to consistent self-improvement.
3. We show that the rate of self-improvement can be exponential and pretrained models can achieve faster acceleration in easy-to-hard generalization.
4. We deep-dive into reverse addition and investigate **error avalanche** caused by label noise, a key failure case of the self-improvement process.

## 2. Related Works

**Length and Easy-to-Hard Generalization.** Length generalization is concerned with extrapolating to longer sequence lengths than those seen during training (Anil et al., 2022). Previous approaches to improve length generalization includes architectural modifications, including specialized positional embeddings (Li et al., 2023; Ruoss et al.,

2023; McLeish et al., 2024; Kazemnejad et al., 2024; Sabbaghi et al., 2024; Cho et al., 2024; Zhou et al., 2024), looping (Fan et al., 2024), novel attention mechanisms (Duan et al., 2023), and input format augmentation (Zhou et al., 2023; 2024). In contrast, our approach adheres to the standard transformer architecture without introducing significant modifications to architecture, positional encoding, or input structure. While prior approaches typically rely on one fixed-length training dataset, we alternate between training and generating training datasets.

More generally, easy-to-hard generalization is the paradigm where human annotation is provided for easier tasks but aiming to enable generalization to harder tasks with no additional supervision (Schwarzschild et al., 2021; Bansal et al., 2022; Burns et al., 2023; Hase et al., 2024; Sun et al., 2024). For instance, Zhang et al. (2024) study this *transcendence* phenomenon in chess, showing that chess transformers can sometimes outperform all players in the training dataset. Similarly, Sun et al. (2024) finds that a reward model trained on easier math problems can be effectively transferred to harder problems, through reinforcement learning.

**Self Improvement.** When high-quality training labels are unavailable or costly to obtain, training on self-generated labels provides an efficient way to enhance model capabilities. Typically, this involves generating candidate labels, filtering or verifying them to prune errors, and retraining on the refined self-generated data (Zelikman et al., 2022; Wang et al., 2022b; Huang et al., 2022; Singh et al., 2023; Chen et al., 2023; Gulcehre et al., 2023; Madaan et al., 2024; Yuan et al., 2024; Liang et al., 2024). This approach has been successfully applied across various domains, including reasoning (Zelikman et al., 2022; Huang et al., 2022; Singh et al., 2023), mathematics (Zhang & Parkes, 2023; Charton

et al., 2024; Liang et al., 2024), coding (Chen et al., 2023), and general instruction tuning (Wang et al., 2022b; Yuan et al., 2024).

Extensive discussion of related works is in Appendix A.

## 3. Preliminaries and Experimental Setup

In this section, we describe the experimental setup, including the model architecture, tasks, training methodology, evaluation criteria, and the self-improvement framework.

**Models** We adopt the LLaMA architecture with six layers, six attention heads, and an embedding dimension of 384 and 14m parameters. We modified the architecture to use No Positional Encoding (NoPE) proposed in (Kazemnejad et al., 2024). We use character-level tokenization across all tasks except for the maze-solving task, where numbers ranging from 0 to 99 are tokenized as individual tokens.

**Tasks** We evaluate our approach on a diverse set of tasks, categorized into arithmetic operations, string manipulation, and maze solving. All tasks we consider admit a straightforward notion of difficulty. We denote the difficulty level of a problem instance $x$ as an integer $\text{Difficulty}(x)$. Table 1 provides examples, difficulty definitions, and relevant sections of each task.

- **Arithmetic operations:**
  1. *Addition* : We consider both reverse and forward addition of two numbers of equal length. In reverse addition, both operands and the answers are reversed, so they are written with the least significant digit first. Forward addition, in contrast, follows the standard format, with the most significant digit first.
  2. *Multiplication* : Multiplication tasks are presented in a chain-of-thought (CoT) data format (Deng et al., 2024), which includes intermediate steps to guide the computation.
- **String manipulation:**
  1. *Copy* : Copying the input sequence.
  2. *Reverse* : Reversing the input sequence.
- **Maze solving:** The task is to solve mazes represented as tree graphs. Given a tree graph and a specified start node and end node, the goal is to find the shortest path.

**Data Generation and Sampling** We generate an initial supervised training dataset $\mathcal{D}_0$ of up to a fixed difficulty level $d_0$ by uniformly sampling the difficulty level $d \leq d_0$, followed by independent sampling of the data conditioned on the difficulty. Denoting the input as $x_i$, labels as $y_i$,

$$\mathcal{D}_0 = \{(x_i, y_i)\}_{i=1}^{N_0}, \quad \text{where Difficulty}(x_i) \leq d_0.$$

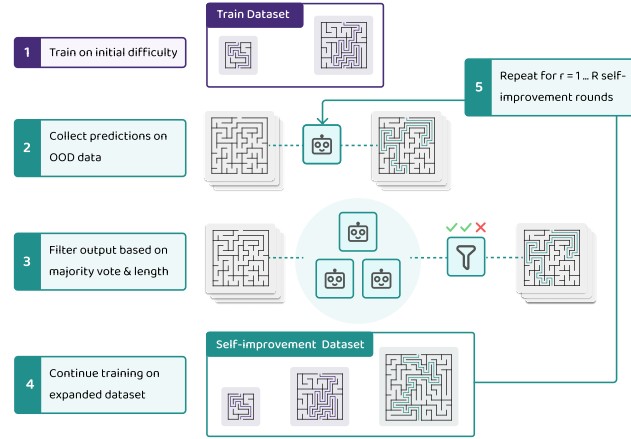

*Figure 2.* Illustration of our self-improvement procedure. At each round, the training data is updated with the model's predictions on progressively harder problems.

Details on data generation and sampling are provided in Appendix C.2.

**Self-Improvement Framework** The self-improvement framework begins by training a model using the labeled training dataset $\mathcal{D}_0$, which gives us our base model $M_0$.

For each subsequent round $r$ ($r = 1, 2, 3, \dots$), we increase the problem difficulty, such as the number of digits or string length for arithmetic and string manipulation tasks, or the number of hops for maze-solving tasks, to $d_r$. Using the previous model $M_{r-1}$, we generate $N_r$ new self-improve data samples $\mathcal{D}_r$ defined as:

$$\mathcal{D}_r = \{(x_i, M_{r-1}(x_i))\}_{i=1}^{N_r}, \quad d_{r-1} \leq \text{Difficulty}(x_i) \leq d_r$$

Instead of the true labels $y_i$, we obtain the predicted labels $M_{r-1}(x_i)$ from the output of the model.

At each self-improvement round $r$, the model is trained on the combined dataset $\mathcal{D}_0 \cup \mathcal{D}_1 \cup \cdots \cup \mathcal{D}_{r-1}$, which includes the initial labeled dataset and all subsequent self-improvement datasets. To ensure sufficient training on the most recently generated data $\mathcal{D}_{r-1}$, we up-sample it with a sampling probability of 50%. The remaining datasets $\mathcal{D}_0, \dots, \mathcal{D}_{r-2}$ are sampled uniformly at random. This iterative process allows the model to gradually tackle harder problems, leveraging its own predictions to expand the training data and improve generalization.

**Data Filtering** We employ two unsupervised data-filtering methods to refine our self-improvement dataset: 1) length filtering and 2) majority voting. For a given self-improved dataset $\mathcal{D}_r = \{(x_i, M_{r-1}(x_i))\}_{i=1}^{N_r}$ at round $r$, data is filtered based on specific criteria on the model-generated outputs $M_{r-1}(x_i)$, producing a smaller, refined dataset $\tilde{\mathcal{D}}_r = \{(x_i, M_{r-1}(x_i))\}_{i=1}^{\tilde{N}_r}$. We provide more details on the motivation and implementation in Section 5.

*Table 1.* Examples of Tasks Considered

| Task Type | Input (Q: Prompt, A: label) | | Task Difficulty | Sections |
|---|---|---|---|---|
| Reverse Addition | Q: 31558+91786=  A: 232451 | | Max digit length of the two operands | 4.1 |
| Forward Addition | Q: 85513+68719=  A: 154232 | | | 6.1 |
| Multiplication | Q: 34895*148=  A: 348950+0273932(3653542)+00447874=36972305 | | | 6.2 |
| Copy | Q: 12345=  A:12345 | | Length of string | 4.2 |
| Reverse | Q: 12345=  A: 54321 | | | |
| Maze Solving | 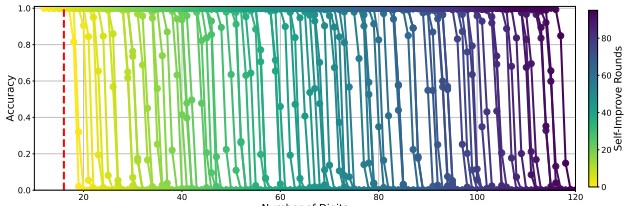 | Finding shortest path from node 2 to 19 (← example image for illustration)  Q: 2>19#73:70,75-97:2,70-70:73,97,59 -75:73,30,19-2:97-30:75-59:70-19:75=  A: 2>97>70>73>75>19 | (1) Number of hops between start & end  (2) Number of nodes | 6.3 |

**Training and Evaluation** Except for the experiments on pretrained Llama 3.2 models, all models are trained from scratch using the conventional next-token prediction objective. The loss is computed solely on the completion, meaning that the input prompt is masked, and only the model's predictions are included in the loss computation. Detailed settings, including hyperparameters and training schedules, are provided in the Appendix C.3.

During inference, we use greedy decoding and exact-match accuracy as the primary metric for evaluation. A prediction is deemed correct if all tokens in the output sequence match the ground truth; any discrepancy in the generated tokens is classified as an incorrect prediction.

# 4. Length Generalization on Reverse Addition and String Copying / Reversing

In this section, we apply our self-improvement framework to reverse addition and string copying/reversing.

## 4.1. Reverse Addition

Reversed addition, where the operands and output are written with the least significant digit first, has been shown to enhance sample efficiency and performance (Lee et al., 2023). Reversed addition has become a popular setting for studying length generalization in arithmetic tasks (Lee et al., 2023; Shen et al., 2023; Zhou et al., 2023; 2024; Cho et al., 2024; McLeish et al., 2024).

**Results** Figure 3 demonstrates that, starting with a model trained on 1 to 16-digit reverse addition, the self-improvement framework enables near-perfect length generalization up to 100 digits without any additional supervision or modifications to positional encodings, input formats, or the Transformer architecture.

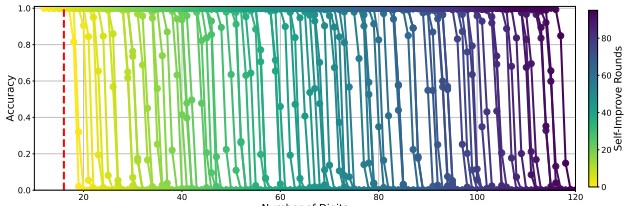

*Figure 3.* Reverse addition task. The self-improvement framework enables a model initially trained on 1-16 digit examples to generalize perfectly to over 100-digit addition. Each shade of color is a different self-improvement round.

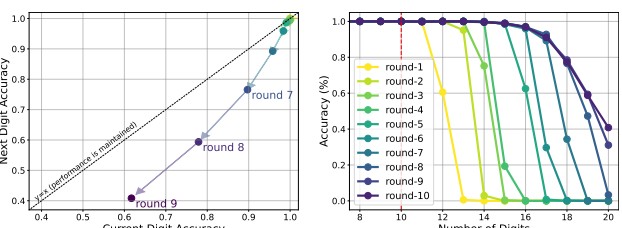

*Figure 4. Error avalanche* is a common failure case for self-improvement. As inaccuracies in self-generated data accumulate, they degrade future rounds of training, leading to eventual failure.

## 4.2. String Copy & String Reverse

Copying and reversing input string is another task that is considered hard for vanilla transformers (Zhou et al., 2023).

**Results.** Similar to reverse addition task, Figure 31 demonstrates that starting with strings of length 1 to 10, the self-improvement framework enables the model to perfectly generalize to string lengths of over 120 after approximately 100 self-improvement rounds.

# 5. Unsupervised Data Filtering

Our framework leverages models' ability to generalize slightly beyond their training difficulty to sample increasingly hard examples. A critical component for success is

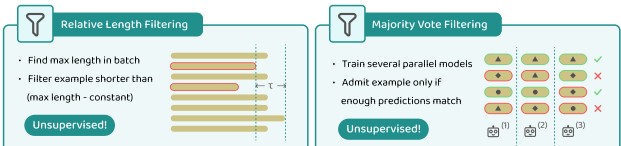

Figure 5. Overview of the two data-filtering methods employed. *Length filtering* removes data points with outputs shorter than a predefined threshold, relative to the maximum output length in the batch. *Majority voting* filters data based on consensus among predictions from multiple models trained on different seeds.

the quality of the self-generated data. Low-quality data can negatively impact the model's generalization performance, leading to even lower-quality data in subsequent rounds and ultimately causing a cascading degradation of the self-improvement process as illustrated in Figure 4.

While cascading error effects are analyzed in greater detail in Section 8 and Appendix 5, this section focuses on two key data-filtering methods used in this work: length filtering and majority voting (Figure 5). And in Section 6 we apply the filtering methods to enable difficulty generalization in forward addition, multiplication and mazes.

**Relative Length Filtering.** A common error in model-generated data is that the generated labels are often shorter than the correct answers (Figure 17). These observations motivate a filtering method based on the relative lengths of model-generated predictions. Specifically, predictions shorter than a predefined threshold—calculated relative to the maximum prediction length within their batch—are filtered out. For a batch of model-predicted outputs, we identify the maximum length of the output $L = \max |M_{r-1}(x_i)|$ and filter out predictions $M_{r-1}(x_i)$ with lengths shorter than a predefined threshold $\tau$. This method is *unsupervised*, as it relies solely on comparing lengths within model-generated outputs rather than referencing ground-truth labels. While particularly suited to length generalization tasks, where harder problems are expected to yield longer answers, length-based filtering shows broader potential for addressing similar challenges in other tasks.

**Majority Voting** Generating multiple candidate answers to ensure self-consistency is a widely used approach for enhancing data quality (Huang et al., 2022; Wang et al., 2022a; Qu et al., 2024; Peng et al., 2024). However, unlike the common practice of sampling multiple reasoning paths by generating outputs with a non-zero temperature, our task of interest requires a single correct answer for each instance. To address this, we train $k$ models $(M_{r-1}^{(1)}, \cdots, M_{r-1}^{(k)})$ using different random seeds and self-improvement data, then apply a majority-voting mechanism with a threshold $\tau$.

Concretely, for each self-improved dataset $\mathcal{D}_r^s = \{(x_i, M_{r-1}^{(s)}(x_i))\}_{i=1}^{N_r}$ where $s$ is the seed index, we fil-

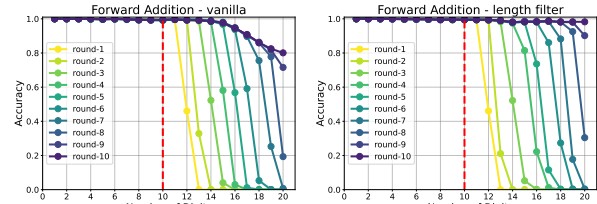

Figure 6. Models trained on forward addition over 10 self-improvement rounds. (Left) Without data filtering. (Right) With length-based filtering using a threshold of 2. Data filtering significantly enhances length generalization performance.

ter the data such that only pairs $\{(x_i, M_{r-1}^{(s)}(x_i))\}$ where $M_{r-1}^{(s)}(x_i)$ matches at least $\lceil \tau \times k \rceil$ outputs among the $k$ models are retained. This ensures that only high-consensus data are preserved for training in subsequent rounds, thereby significantly improving overall data quality and model performance. This approach is conceptually similar to an iterative version of the bagging algorithm (Breiman, 1996).

# 6. Length and Difficulty Generalization on Forward Addition, Multiplication, Maze

We extend our evaluation to a class of harder tasks, including forward addition, multiplication, and maze-solving. Our results demonstrate that the framework is not limited to length generalization but extends to **difficulty generalization**, where the model incrementally learns to solve increasingly difficult problems. By employing controlled sampling of problem difficulty and data filtering techniques for each round, the model successfully adapts to harder tasks, highlighting the versatility and robustness of the self-improvement approach.

## 6.1. Forward Addition

Forward addition is a straightforward task, yet very challenging for transformer models to length generalize on. In reverse addition, each step only requires processing a fixed-size subset of the input. However, in the forward addition, the size of the relevant input required to generate correct tokens increases, making the problem more complex (Zhou et al., 2023).

**Results.** Figure 6 shows the results of forward addition experiments, where the model is initially trained on labeled data of up to 10 digits and then undergoes 10 rounds of self-improvement. Without any data filtering (Left), the model's performance begins to deteriorate after a few rounds of training, leading to a collapse in generalization. However, applying the length-based filtering approach with a threshold length of 2 results in significant improvements in length generalization performance (Right). By refining the self-improvement dataset at each round, the self-improvement

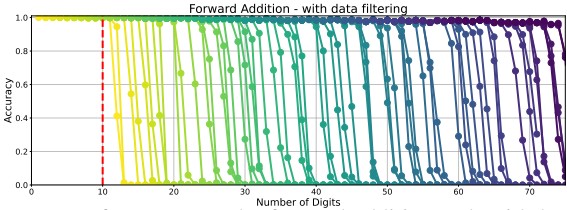

*Figure 7.* Performance on the forward addition task with length filtering. The model is initially trained on labeled forward addition data of lengths 1 to 10. With over 60 self-improvement rounds, the model achieves strong generalization to lengths up to 75.

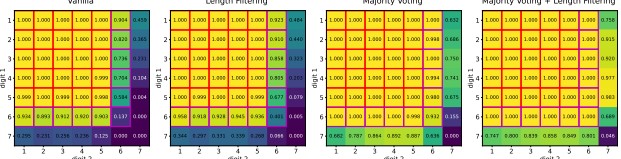

*Figure 8.* Comparison of filtering methods at round 7. From left to right: no filtering, length filtering, majority voting, and a combination of majority voting and length filtering. Data filtering significantly improves self-improvement performance, with the combined approach achieving the best results.

framework remains robust across multiple rounds.

With continued training over 60 self-improvement rounds, the model maintains performance exceeding 98% accuracy for sequences up to length 70 (Figure 7). This demonstrates the effectiveness of length-based filtering in sustaining the self-improvement process and enabling models to generalize to much longer sequences.

## 6.2. Multiplication

We also extend our approach on multiplication, which is a challenging task even in-distribution (Dziri et al., 2024). Fine-tuning large language models on datasets with chain-of-thought(CoT) steps has shown limited success. We adopt a data format similar to Deng et al. (2024), where multiplication is given a problem of multiplying two numbers, the label expands the multiplication into steps that include partial products of multiplying the first operand with each digit of the second operand and the intermediate results.

The model is initially trained on $n$-by-$n$ multiplication examples with $n = 5$. Directly introducing $n+1$-by-$n+1$ examples results in poor performance, hence, we adopt a more fine-grained difficulty schedule where we sample $n+1$-by-$m$ and $m$-by-$n+1$ examples with $m$ growing from 1 to $n+1$. This gradual progression allows the model to adapt incrementally to larger operand sizes, making the transition to harder examples more manageable.

**Results.** To improve the quality of self-generated training data, we apply three data filtering methods: length filtering, majority voting, and a combination of both (Appendix C.3).

Figure 8 compares the effectiveness of these filtering meth-

ods at round 7, where models are trained on self-generated data for up to 6-by-6 multiplication. All three filtering methods enhance self-improvement, with majority voting outperforming length filtering. The combined approach—applying both majority voting and length filtering—achieves near-perfect generalization to 6-by-6 multiplication.

Training for additional rounds further extends this generalization. The combined filtering strategy continues to yield near-perfect accuracy up to 9-by-9 multiplication at round 31 (Figure 36), with the potential for even further generalization in subsequent rounds. We further demonstrate that we can accelerate the process, achieving perfect performance on 10-by-10 multiplication in just 19 rounds (Figure 23).

## 6.3. Maze

We extend our evaluation from arithmetic to a more complex problem: finding the shortest path in a maze. Pathfinding presents significant challenges for autoregressive models (Bachmann & Nagarajan, 2024). Our mazes can be represented by a tree graph in a 2-dimensional space and they do not have loops. Figure 30 provides a visualization of this task and the corresponding input and output data format. Details on maze generation are provided in Appendix C.2.3.

We evaluate two generalization settings: 1) increasing the number of hops while keeping the number of nodes fixed, and 2) increasing the number of nodes while keeping the number of hops fixed. In the first setting, the input graph description remains constant in size, but the output length grows as the difficulty increases. In the second setting, the input graph expands with more nodes, while the output remains of fixed length.

### 6.3.1. INCREASING THE NUMBER OF HOPS

The difficulty of the maze-solving task increases with the number of hops required from the start node to the end node. We begin by training the model on a labeled dataset containing paths of up to $h = 9$ hops. In each self-improvement round, we increase $h$ by one, progressively introducing longer paths, while fixing the number of nodes $N = 30$.

**Results.** As shown in Figure 9, without data refinement, self-generated training data degrades over successive rounds, leading to an eventual collapse in the self-improvement process. In contrast, majority voting stabilizes data quality, allowing near-perfect data quality and the model continues to successfully generalize to paths up to 30 hops.

### 6.3.2. INCREASING THE NUMBER OF NODES

Another approach to increasing task difficulty is to expand the number of nodes in the graph while keeping the number of hops fixed at $h = 9$. We begin by training the model on

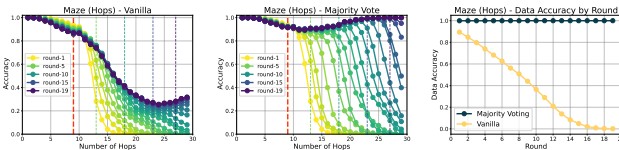

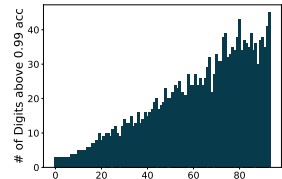

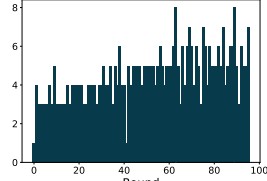

*Figure 11.* Number of **extra** OOD digit lengths achieving over 99% accuracy when self-improving with one additional digit per round, on (Left) copy and (Right) reverse addition tasks. The growing OOD capability suggests the potential to sample more digits per round as self-improvement progresses.

*Figure 9.* Maze-solving with increasing hops ($N = 30$ nodes). Models are trained on graphs with up to 9 hops and generalized by incrementally increasing hops by 1 in each self-improvement round. Results show mean accuracy across 3 seeds. (Left) No filtering. (Middle) Majority voting. (Right) Self-improve data accuracy per round. Filtering significantly enhances data accuracy and improves generalization.

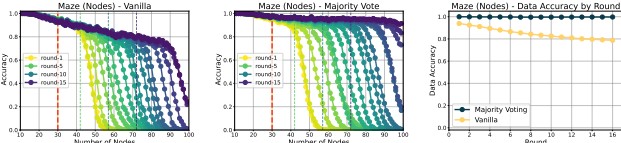

*Figure 10.* Maze-solving with increasing nodes ($h = 9$ hops). Models are trained on graphs with up to 30 nodes and generalized by incrementally increasing the number of nodes by 3 per round. Majority voting improves generalization to larger graphs.

a labeled dataset containing paths of fixed number of hops $h = 9$, and nodes $N = 10$ to 30. In each self-improvement round, the number of nodes is increased by 3.

**Results.** As shown in Figure 10, training without filtering leads to gradual performance degradation, whereas majority voting preserves high-quality data, maintaining a self-improvement accuracy above 99.7% and enabling generalization to larger graphs with 9 hops.

While these experiments focus on fixing one dimension (number of hops or number of nodes) and increasing the other, alternating between increasing the difficulty in both dimensions is expected to generalize the maze-solving task to handle larger graphs and longer paths simultaneously.

### 6.3.3. VERIFIERS.

Solving the shortest path problem can be computationally expensive, but verifying the correctness of a given solution is significantly simpler. A valid path can be verified by traversing the sequence and ensuring three conditions: 1) each move is valid, meaning the path follows adjacency constraints; 2) the final destination matches the intended goal; and 3) no nodes are repeated, confirming that the solution is indeed the shortest path.

We show in Appendix B.3 that while filtering self-generated data using oracle verifiers based on move and end validity is effective, majority voting based filtering—without any external verification—performs comparably, highlighting its effectiveness.

## 7. Ablations

### 7.1. Increasing OOD generalization with more self-improvement

Sampling instances that are too difficult for the current model is detrimental to the quality of self-improvement data, which causes downstream performance to break down. However, in tasks like reverse addition and copy, we observe that the out-of-distribution (OOD) extrapolation capabilities improve progressively as the model undergoes more rounds of self-improvement, which means we can sample more and more difficulty levels every round. Figure 11 illustrates how the number of additional OOD lengths achieving over 99% accuracy grow with each round when the model is self-improved using only one additional digit per round. The model's OOD extrapolation capabilities expand as it is trained on longer sequences.

### 7.2. Accelerating self-improvement

Since the amount of extra OOD generalization increases roughly linearly with each additional round of self-improvement (Figure 11), sampling as many difficulty levels as possible per round could lead to exponential improvements in performance. Therefore, we propose an accelerated self-improvement schedule: At each round, the self-improvement dataset is uniformly sampled from *all difficulty levels achieving over 99% evaluation accuracy*, instead of incrementally sampling by only one additional length. As shown in Figure 12, this approach allows the model to achieve 100 digit extrapolation with less than half of the rounds. All other hyperparameters remain unchanged. We also provide results in the multiplication setting in Figure 23.

### 7.3. Pretrained Models

We extend our self-improvement framework to pretrained models, specifically Llama-1B and Llama-3B (AI@Meta, 2024), to explore scaling effects and the impact of finetuning on larger models. For consistency in tokenization, we use character-level tokenization instead of the default tokenizer of the Llama models, and use LoRA (Hu et al., 2021).

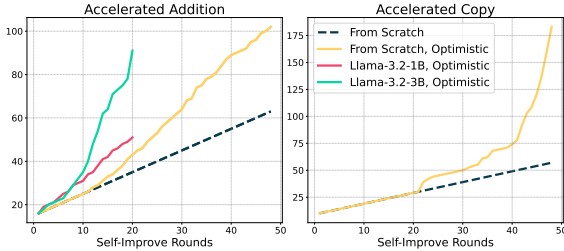

*Figure 12.* Maximum input length achieving over 99% accuracy at different self-improvement rounds for (Left) Reverse addition and (Right) Copy task. The dashed linear line represents the standard schedule of sampling one additional length per round. Faster self-improvement schedules allow the model to generalize to longer inputs with fewer rounds. Furthermore, finetuning from pretrained models enhances the acceleration.

**Results.** Larger models achieve better extrapolation performance, which leads to faster acceleration with larger models. Figure 12 compares self-improvement acceleration between Llama-3B, Llama-1B, and a smaller 14M parameter model trained from scratch. The results demonstrate that larger pretrained models can generalize to longer sequences with fewer rounds of self-improvement.

## 8. Error Avalanches in Self-Improvement

The success of self-improvement hinges on the accuracy of self-generated data. Figure 4 highlights a key challenge for out-of-distribution (OOD) generalization, specifically in $n + 1$-digit performance. Inaccuracies in self-generated $n$-digit data negatively affect generalization, leading to even poorer performance in the subsequent round.

This cascading effect, or *error avalanche*, compounds over successive self-improvement rounds, ultimately degrading the model's overall performance and risking a collapse of the self-improvement process.

A natural question is: *how much error is required to trigger an avalanche?* We investigate this question by first characterizing the model mistakes, and then injecting synthetic wrong examples into the self-improvement data.

**Simulating Avalanche** We identify that model mistakes are mainly of two types: (1) incorrect digits by $\pm 1$ and (2) dropped digits (see section B.6). We simulate these errors by constructing four kinds of noises:

- uniform: replacing the label with another random number of the same length.
- perturb: perturbing last three digits of the label by $\pm 1$.
- drop-digits: dropping 1-3 digits from the last 3 digits.
- drop-perturb: first applying "perturb" and then "drop-digits", effectively combining the effect of both noises.

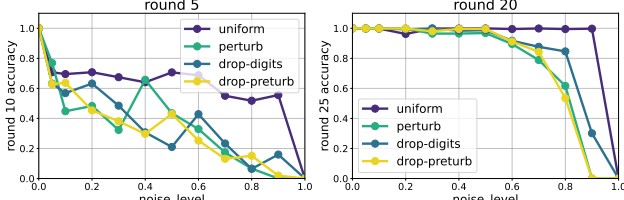

*Figure 13.* Simulating error avalanche. Errors are injected at the end of rounds 5 and 20, and the self-improvement process continues for five additional rounds. The model tolerates errors up to a threshold before performance collapses. This tolerance increases with more rounds of self-improvement.

We inject these errors at rounds 5 and 20 of the reverse addition task and track their effects after five subsequent self-improvement rounds. As shown in Figure 13, injecting enough of these errors into the training data causes performance on the next difficulty to crash. In particular, we find that 1) structured noise (perturb, drop-digits) are more harmful than uniform noise and 2) more rounds of self-improvement lead to robustness against label noise.

These findings emphasize the critical need for maintaining high-quality self-generated data to sustain effective and persistent self-improvement. We present more results on label noise and robustness in Section B.7

## 9. Limitations

In our framework, we identify three core limitations: First, our approach only generates solutions (labels) rather than new input instances during self-improvement. It is a separate challenge to model input distributions based on task difficulty. Second, defining and quantifying task difficulty remains an open challenge in real-world domains, though we find models show some robustness to imperfect difficulty scheduling, particularly with pretraining. Finally, while our framework assumes models can handle slightly harder tasks than their training data, this may not hold for all problems—as demonstrated by raw multiplication tasks. However, breaking problems into intermediate steps can enable the necessary out-of-distribution generalization for self-improvement.

## 10. Conclusion

In this work, we have shown self-improvement training enables transformers to gradually generalize from easy to hard problems without access to hard labels. One extension is to incorporate more sophisticated verifiers as well as problem classes that is easy to verify but hard to solve. We expect self-improve to synergize with strong verification to enable transformers to solve harder problems beyond arithmetic or mazes.

## Impact Statement

This paper presents work whose goal is to advance the field of Machine Learning. There are many potential societal consequences of our work, none which we feel must be specifically highlighted here.

## Acknowledgment

D.P. is supported by NSF Medium award No. 2403074, ONR Grant No. N00014-21-1-2806 and No. N00014-23-1-2848. K.L. is supported by NSF Award DMS-2023239, NSF CAREER Award CCF-2339978, Amazon Research Award, and a grant from FuriosaAI.

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

# A. Detailed Discussion of Related Work

**Length Generalization.** While Transformers (Vaswani et al., 2017) have achieved remarkable success, they often struggle with length generalization—where a model trained on problems of fixed length fails to extrapolate to longer sequences (Dubois et al., 2019; Hupkes et al., 2020; Newman et al., 2020; Anil et al., 2022). Addressing this limitation is crucial, as poor length generalization indicates that language models may not fully understand the underlying task. Zhou et al. (2023) hypothesize that Transformers are more likely to length generalize on tasks with small RASP-L complexity. They demonstrate that tasks such as reverse addition and copying have low RASP-L complexity, making them easier to length generalize, whereas forward addition poses a greater challenge.

Several approaches have been proposed to improve length generalization, particularly in arithmetic tasks. These include modifications to positional embeddings, such as Abacus embeddings (McLeish et al., 2024), NoPE (Kazemnejad et al., 2024), FIRE (Li et al., 2023), and pairwise positional encodings (Sabbaghi et al., 2024; Cho et al., 2024), as well as randomized positional encodings (Ruoss et al., 2023; Zhou et al., 2024). Other approaches focus on architectural changes, such as introducing looping mechanisms (Fan et al., 2024) or incorporating hand-crafted bias corrections in attention score matrices (Duan et al., 2023). Additionally, input modifications, such as index hinting, have been explored to enhance generalization (Zhou et al., 2023; 2024).

Beyond arithmetic, length generalization has also been studied in broader contexts. For instance, size generalization in graph-based tasks has been investigated (Yehudai et al., 2021), while Kim et al. (2024) leverage a cognitive map framework for path planning to tackle the maze-solving problem.

In contrast, our approach adheres to the standard transformer architecture without introducing modifications to architecture, positional encodings, or input structure. A key distinction lies in the training methodology. While prior approaches typically rely on fixed-length training datasets without further updates to model weights, we iteratively update model weights on self-generated datasets, enabling the model to progressively improve and extend its generalization capabilities.

Our self-improvement framework and results on forward addition (Section 6.1) are closely related to those of Zhang & Parkes (2023), where self-training enables forward addition generalization from 6-digit examples to 24-digit addition. Like their approach, we iteratively apply self-training on progressively harder problems. However, a key distinction is that their method follows a two-step process in each round: first generating solutions using chain-of-thought (CoT) reasoning, then fine-tuning the model to produce direct answers without CoT.

Our multiplication results in Section 6.2 have relevance with findings by Jelassi et al. (2023), who showed that dataset priming (adding a small number of labeled long-sequence examples) can enable length generalization for multiplication (although this is not strictly out-of-distribution). Our approach of incorporating accurate, self-generated out-of-distribution data via filtering can be seen as an automated form of dataset priming. Furthermore, while our approach uses chain-of-thought (CoT) data for multiplication, we believe it is possible to length generalize on non-COT multiplication as well, by incorporating methods like Deng et al. (2024) to help the model iteratively internalize the CoT steps.

**Easy-to-hard Generalization.** Our self-improvement framework operates in a setting where human annotation is provided for easier tasks, enabling generalization to harder tasks with no additional supervision. This paradigm, often referred to as easy-to-hard generalization (Schwarzschild et al., 2021; Bansal et al., 2022; Burns et al., 2023; Hase et al., 2024; Sun et al., 2024), leverages the transfer of learned policies or reward models from simpler problems to more challenging ones. For instance, Zhang et al. (2024) study this phenomenon in chess, showing that chess transformers can sometimes outperform all players in the training dataset. Similarly, Sun et al. (2024) finds that a reward model trained on easier mathematical problems can be effectively transferred to harder problems, facilitating generalization through reinforcement learning. Shin et al. (2024) identifies overlap data points—instances containing both easy and hard patterns—as a key mechanism for weak-to-strong generalization, allowing weak models to pseudolabel easier patterns while stronger models use these labels to learn harder patterns. Our work shows that a similar mechanism emerges naturally within self-improvement, where progressively increasing difficulty enables models to generate useful supervision signals for harder tasks without explicit human intervention.

**Self Improvement.** Self-training is a common approach in semi-supervised learning that leverages unlabeled data to enhance model performance (Zhu, 2005; Pise & Kulkarni, 2008; Yang et al., 2022). When high quality training labels are not available, training on self-generated labels is an efficient way to extract more capabilities from the model. Usually, this involves generating candidate labels, pruning wrong labels through verification or filtering, and retraining with self-generated

data. ReST (Gulcehre et al., 2023) and I-SHEEP (Liang et al., 2024) propose self-improvement as a general purpose alternative to reinforcement learning from human feedback (RLHF), while Yuan et al. (2024) propose "self-rewarding" model that generates its own instruction tuning set.

The self-improvement framework has been applied to a wide range of tasks. For example, Zhang et al. (2019) replaces an expensive teacher distillation with self-distillation for image recognition tasks. In LLM reasoning domains, Huang et al. (2022); Singh et al. (2023); Pang et al. (2024), and STaR (Zelikman et al., 2022) bootstrap complex reasoning capabilities by asking models to generate rationales for unlabeled questions and training on self-generated rationals that yielded correct answers. Similarly, Zhang & Parkes (2023) shows self-improving using chain-of-thought (COT) data sampled from the model allows generalization of the integer addition task to more digits. For coding tasks, Chen et al. (2023) teaches LLMs to self-debug with feedback using self-generated code explanation and unit test execution results. In mathematics, PatternBoost (Charton et al., 2024) shows that transformers can discover unsolved mathematical constructions of various problems using an algorithm that alternates between sampling constructions from the model (local search) and training on self-generated data (global learning). Similarly, Alfarano et al. (2024) generate synthetic training samples to train transformers to discover new Lyapunov functions. Recent works also investigate theoretical and empirical aspects of self-improvement. Bansal et al. (2024) highlight the effectiveness of smaller models in self-improvement, while Song et al. (2024) identify the generation-verification gap as a key factor governing the self-improvement process. Huang et al. (2024) introduce the "sharpening mechanism," where training on best-of-N responses from the model amortizes maximum likelihood inference and improves output quality.

Our work is greatly inspired by ReST (Gulcehre et al., 2023) and STaR (Zelikman et al., 2022), in which models iteratively generate predictions, filter high-quality responses, and fine-tune on the self-generated dataset.

**Model Collapse** Recent research has extensively investigated the phenomenon of model collapse, where repeated training on a model's own outputs leads to performance degeneration and a loss of the true underlying data distribution (Shumailov et al., 2024; Hataya et al., 2023; de Arcaute et al., 2023; Shumailov et al., 2023; Alemohammad et al., 2023; Briesch et al., 2023).

Shumailov et al. (2024) provide evidence that iterative training on model-generated data, without filtering, results in rapid degeneration and forgetting of the true data distribution. They emphasize the importance of preserving original data sources over time. Similarly, Shumailov et al. (2023) show that the tails of the original content distribution diminish after repeated self-training, while Zhang & Parkes (2023) highlight the error avalanching effect, where errors compound as models are trained on their own generated data.

Despite its apparent inevitability, several strategies have been proposed to mitigate model collapse. Research shows that the risk of collapse diminishes when the initial model closely approximates the true data distribution (Bertrand et al., 2023), or when real data is retained throughout training rather than being fully replaced (Gerstgrasser et al., 2024; Dohmatob et al., 2024; Briesch et al., 2023). Additionally, Gillman et al. (2024); Feng et al. (2024) suggest using reliable verifiers during self-training to ensure high-quality self-generated data, further reducing the likelihood of collapse.

Our approach addresses these challenges by maintaining a core labeled dataset throughout training, consisting of examples of limited length or difficulty. Synthetic data, generated incrementally by the model, is added in a controlled manner. By incorporating unsupervised filtering techniques such as length filtering and majority voting, we ensure the quality of self-generated data. Our framework builds upon prior findings by preserving clean data and selectively incorporating synthetic data.

Additionally, our results in Section 8 align with findings from Rolnick (2017), which demonstrate that deep neural networks are robust to significant label noise in image classification tasks. Additionally, Bayat et al. (2024) recently emphasized that memorization alone does not harm generalization; rather, the combination of memorization with spurious correlations is what undermines learning. Our results suggest that despite memorizing past mistakes, the self-improvement framework remains effective, provided that incorrect samples do not dominate the training distribution.

# B. Additional Results

## B.1. Does the Model Truly Learn Addition?

When the two operands are sampled randomly, the probability of encountering an instance with a carry chain length of $N$ decays exponentially with $N$. Under this sampling strategy, the model may rarely, if ever, see "hard[1]" instances of addition, as illustrated in Figure 14. To address this, we manually construct a test dataset to include at least 500 examples for each maximum cascading carry length. This ensures that the evaluation captures the model's ability to handle harder instances of addition.

The results in Figure 15 show that the model is capable of performing additions with up to 20 cascading carries, even though it has never encountered such cases during training. This demonstrates that the model can generalize to harder instances of addition despite being trained predominantly on easier examples.

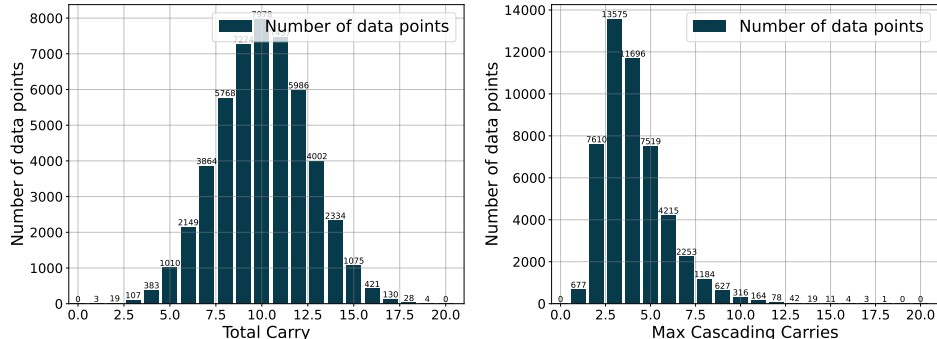

*Figure 14.* Number of carries in the self-improve dataset of 20-digits. The models does not see examples of high numbers of carry during training.

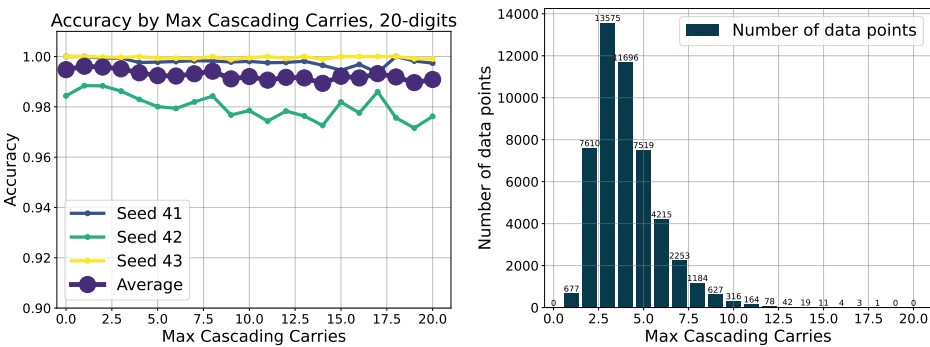

*Figure 15.* Performance of the model at round 10 (trained with self-generated data up to 20 digits). (Left) Accuracy as a function of the maximum cascading carries. (Right) Number of examples with each maximum cascading carry length in the self-improve training dataset. Models can successfully perform hard - with a high number of cascading carries - addition tasks even without encountering such examples in the training dataset.

## B.2. Motivation for Data Filtering

### B.2.1. IMPORTANCE OF DATA FILTERING

Figure 16 demonstrates this effect in the reverse addition task. The x-axis represents the accuracy of the self-improve dataset $\mathcal{D}_r$, generated by model $M_{r-1}$ at round $r$, while the y-axis shows the resulting $n + 1$-digit performance of model $M_r$. The prevalence of data points below the $y = x$ line indicates that low-quality data diminishes performance, underscoring the need for maintaining high-quality data throughout the self-improvement process.

---

[1]we define hard instance of addition to be cases with multiple numbers of cascading carries (Quirke & Barez, 2023)

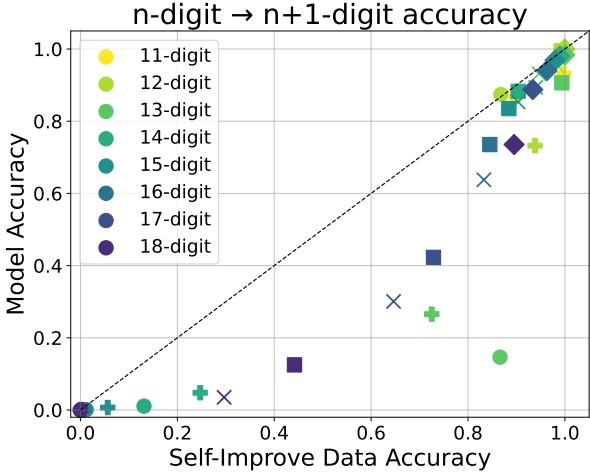

*Figure 16.* Effect of self-generated data accuracy on length generalization performance in the reverse addition task. Each data point represents the accuracy of the self-improve data $\mathcal{D}_r$ (on $n$ digit addition) generated by model $M_{r-1}$, and the resulting $n+1$-digit performance of the trained model $M_r$ at round $r$. The prevalence of points below the $y = x$ line highlights the critical importance of high-quality data for successful self-improvement.

### B.2.2. OOD RESULTS ARE OFTEN SHORT

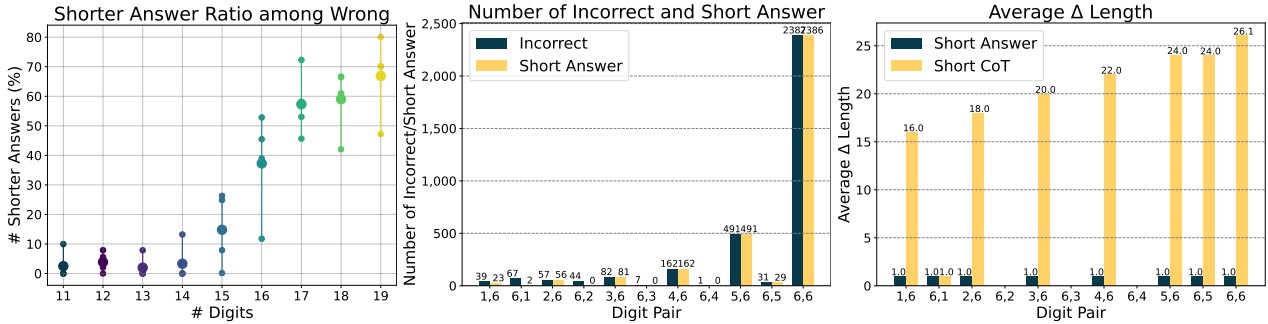

*Figure 17.* (Left) Reverse addition task: the proportion of shorter answers among incorrect predictions increases with each round. (Mid & Right) CoT-multiplication task with majority voting: (Mid) The majority of incorrect answers are short. (Right) The average length discrepancy of short answers compared to the correct answer or the CoT reasoning part.

Figure 17 illustrates this phenomenon for both the reverse addition and CoT-multiplication tasks. In reverse addition (Left), as the number of digits in the training data increases (or as self-improvement rounds progress), the proportion of incorrect self-generated data where the answer is shorter than the correct label length also increases. Similarly, for CoT-multiplication (Mid and Right), most incorrect answers are shorter than the correct ones. Furthermore, in cases where the answers are shorter, the outputs often miss one or more reasoning steps in the chain-of-thought (CoT) reasoning process.

### B.2.3. MAJORITY VOTING LEVERAGES LABEL DIVERSITY

Self-improvement relies on the model's ability to generalize to slightly harder problems. However, this generalization is not always robust and can vary significantly across different training instances (Zhou et al., 2024). Majority voting mitigates this variability by aggregating predictions across multiple independently trained models, thereby improving the reliability of self-generated labels.

To illustrate this variability, Figure 18 shows test accuracy across five models trained with different random seeds on the initial training dataset containing up to 5-by-5 multiplication. Even when trained on identical training data, models exhibit substantial performance differences in extrapolation. Similarly, Figure 19 demonstrates that this variability persists even when models are trained from the same seed data.

Figure 20 demonstrates the effectiveness of majority voting in the multiplication task across five models trained with

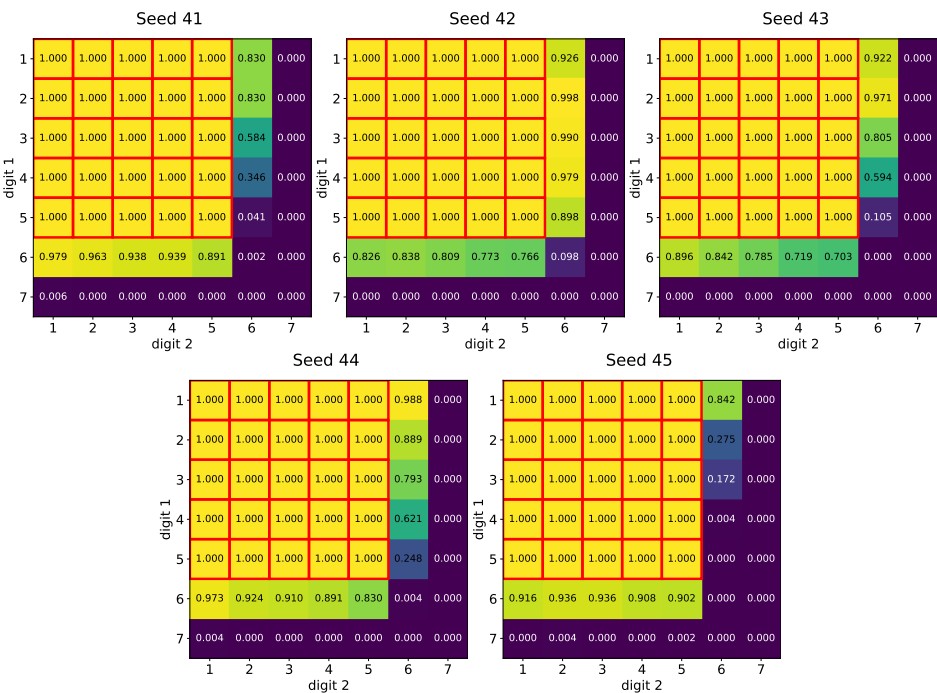

*Figure 18.* Test accuracy on 5 different seeds during the initial training phase. Models exhibit high variance in performance.

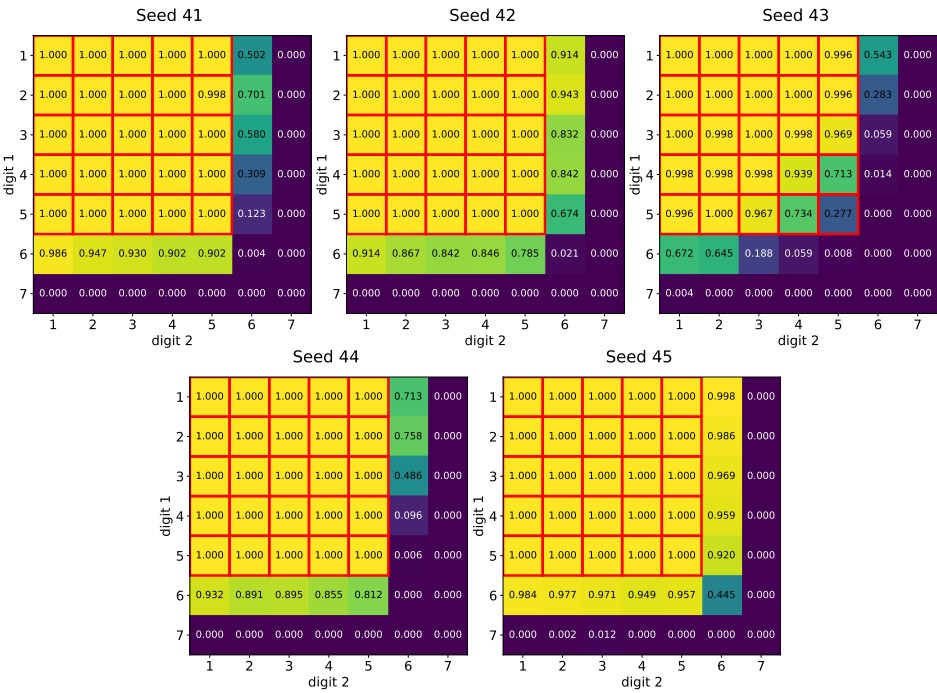

*Figure 19.* Test accuracy on models trained with the same seed data but different training seeds. Despite identical training data, models exhibit large variability.

different seeds during the initial training phase on data $\mathcal{D}_0$, which consists of up to 5-by-5 multiplication problems. The mean accuracy (Left) is relatively low, with a high standard deviation (Mid), indicating substantial variability among the models. By applying majority voting with a consensus on at least 4 out of 5 model outputs, the generated dataset quality improves significantly (Right). For example, while the 5-by-6 multiplication task achieves an average accuracy of 31%

across models, the majority-voting strategy generates a dataset with 93.3% accuracy.

In practice, datasets for larger multiplications, such as 5-by-6 digits, are created after multiple rounds of self-improvement training, gradually incorporating $m$-by-6 and 6-by-$m$ data with incrementally increasing $m$ at each round.

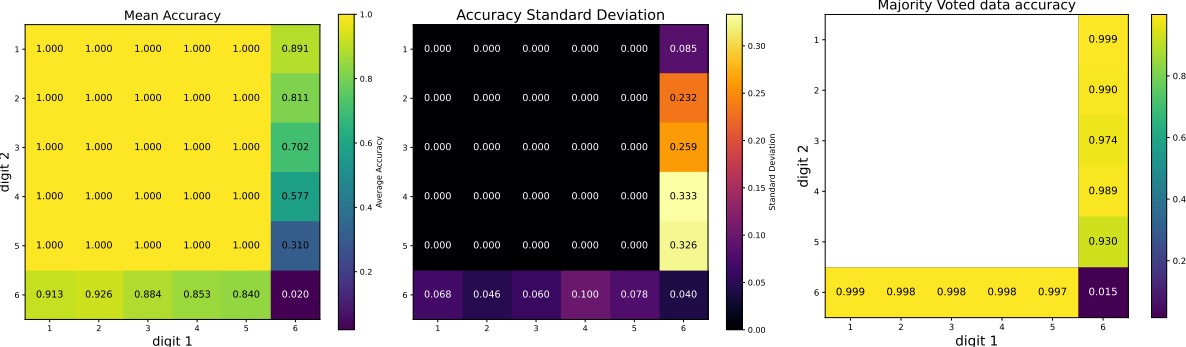

*Figure 20.* (Left & Mid): Mean and standard deviation of accuracy among five models trained with different seeds on the initial training round. (Right): Accuracy of majority-voted data points. Majority voting significantly boosts data quality, with 5-by-6 multiplication data accuracy increasing from an average of 31% to 93.3%

### B.2.4. ABLATIONS FOR MAJORITY VOTING

Our majority voting method requires training multiple models in parallel. In our primary setting, we train $k$ models with different random seeds, allowing each to generate and train on its own independent self-improved dataset at every round.

To evaluate the necessity of training multiple independent models and generating separate self-improvement datasets, we compare our approach against the following baselines:

1. No majority voting, but larger self-improve data: Instead of using multiple models, we train a single model while sampling $k$ times more self-improve data per round, ensuring that the total amount of generated data matches our main setting.

2. Shared self-improve data: We train $k$ models with different initial seeds but subsequently train all models on the same self-improved dataset.

3. Shared initial training seed: All models are initialized from the same seed but then trained on separate self-improved datasets.

4. Our main setting: Each model is initialized with a different seed and trained on its own independently generated self-improve dataset.

Figure 21 presents the performance of these variations, highlighting the importance of training on independently generated self-improve datasets rather than simply increasing dataset size or sharing training trajectories across models.

*Table 2.* Comparison of Data Cost Across Majority Voting Variants

| Method | Initial Training Data Cost | Self-Improve Data Cost (Per Round) |
|---|---|---|
| No Majority Voting, Larger Data | 1 | $k$ |
| Shared Self-Improve Data | $k$ | 1 |
| Shared Initial Training Seed | 1 | $k$ |
| Full Majority Voting (Ours) | $k$ | $k$ |

We set $k = 5$ and report the average performance across five models. Figure 21 shows that simply increasing the amount of self-improvement data without filtering leads to poor performance. Surprisingly, using $5\times$ more self-improvement data per round performs even worse than using less data (Figure 32), consistent with our findings in Section B.7.1.

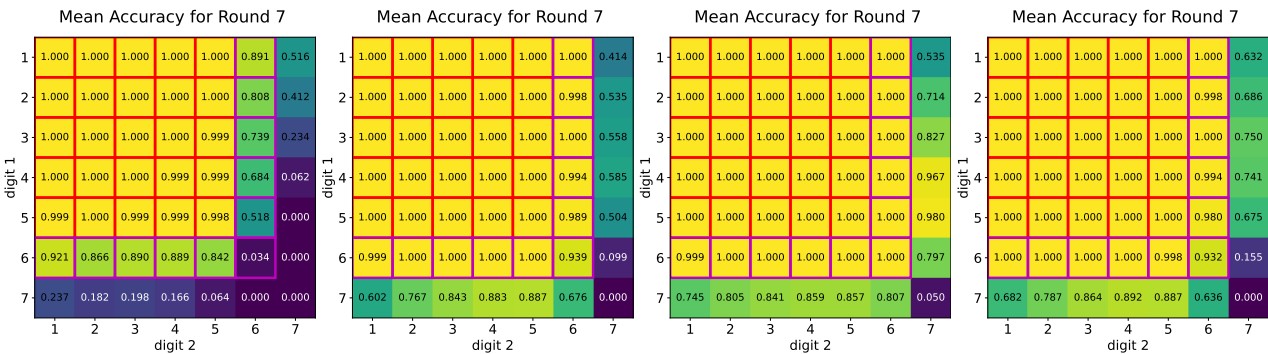

*Figure 21.* Ablations on majority voting. (Left) No majority voting, but larger self-improve data. (Left-Center) Majority voting with shared self-improve data. (Right-Center) Majority voting with shared initial training seed. (Right) Our primary setting with fully independent training and self-improve datasets.

Additionally, majority voting with shared self-improve data (second panel from the left) underperforms in OOD compared to models trained on separate self-improve datasets. This suggests that model diversity—enabled by training on different self-improve data—may be important for majority voting to be effective.

On the other hand, comparing the right two panels in Figure 21, where the difference lies in whether the base models were trained on different labeled data $\mathcal{D}_0$, we find minimal differences in OOD performance. This may be due to the large size of the initial training set (5M examples), which provides sufficient diversity. Furthermore, as Figure 19 shows, models trained on the same initial dataset but with different training seeds still exhibit substantial variability, suggesting that model diversity can emerge from different training trajectories alone.

### B.3. Verification Filters on Mazes

Solving the shortest path problem can be computationally expensive, but verifying the correctness of a given solution is significantly simpler. A valid path can be verified by traversing the sequence and ensuring three conditions: 1) each move is valid, meaning the path follows adjacency constraints; 2) the final destination matches the intended goal; and 3) no nodes are repeated, confirming that the solution is indeed the shortest path.

Self-improvement frameworks commonly incorporate verifiers to filter self-generated data, often leveraging trained models or reward models (Zelikman et al., 2022; Singh et al., 2023; Hosseini et al., 2024; Lightman et al., 2023). While our primary focus is not on training or designing an additional verification mechanism, we investigate the effectiveness of using an external verifier as a data-filtering method.

To this end, we evaluate an oracle verifier that enforces two essential constraints: 1) move validity, ensuring that every transition in the generated solution adheres to the adjacency constraints of the maze, and 2) end validity, confirming that the final node in the solution corresponds to the correct destination. We compare the effectiveness of this oracle-based filtering against self-improvement without data filtering and majority-voting-based filtering to assess its impact on performance and stability.

**Results.** Figure 22 shows results for mazes with increasing hops, increasing nodes, and three different verification strategies: checking moves, checking end validity, and checking both. As expected, verification improves data quality and serves as an effective filtering technique in self-improvement. Notably, verifying move validity proves to be significantly more effective than verifying only the correctness of the end node. Interestingly, however, majority voting—a strategy that does not rely on an external verifier—performs comparably to verification-based filtering. This suggests that self-consistency mechanisms alone can be sufficient for maintaining high-quality training data.

Additional results, including finer-grained analysis of move validity and end validity beyond exact match accuracy, are provided in Appendix D.0.3.

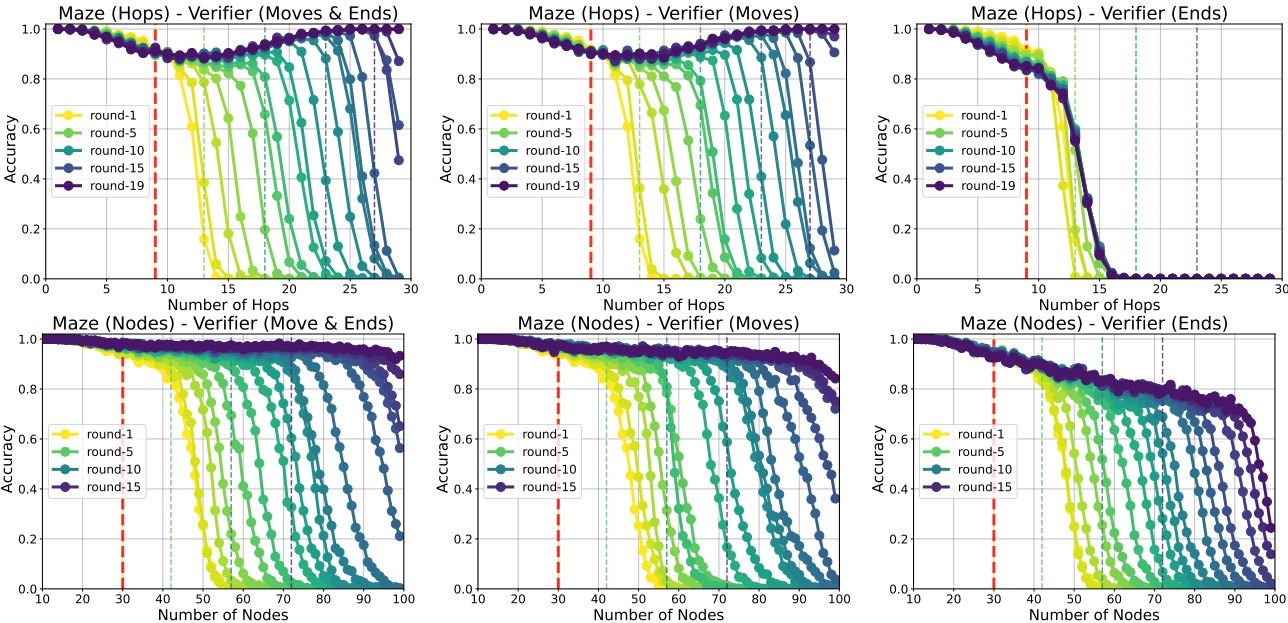

*Figure 22.* (Top) Increasing hops. (Bottom) Increasing nodes. (Left) Verifier on both moves and ends. (Middle) Verifier on moves only. (Right) Verifier on ends only. Verifier-based filtering improves self-improvement performance, with move validation proving more effective than end validation alone. Interestingly, majority voting performs on par with oracle verification, suggesting that self-consistency mechanisms can serve as effective alternatives to explicit verification.

## B.4. Accelerated Self-Improvement for Multiplication

We validate the accelerated self-improvement (Section 7.2) setting to the task of multiplication. For the multiplication task, we observe similar enhancement using an accelerated schedule, as depicted in Figure 23. Under the standard schedule, reaching 10-by-10 multiplication from 5-by-5 requires 41 self-improvement rounds, incrementally increasing one operand by 1 at a time. With the accelerated schedule, we progressively sample more operand pairs as self-improvement proceeds, reducing the required rounds to 19 while achieving perfect test performance (see Figure 37 for full results). The settings for multiplication follow the setting in Section 6.

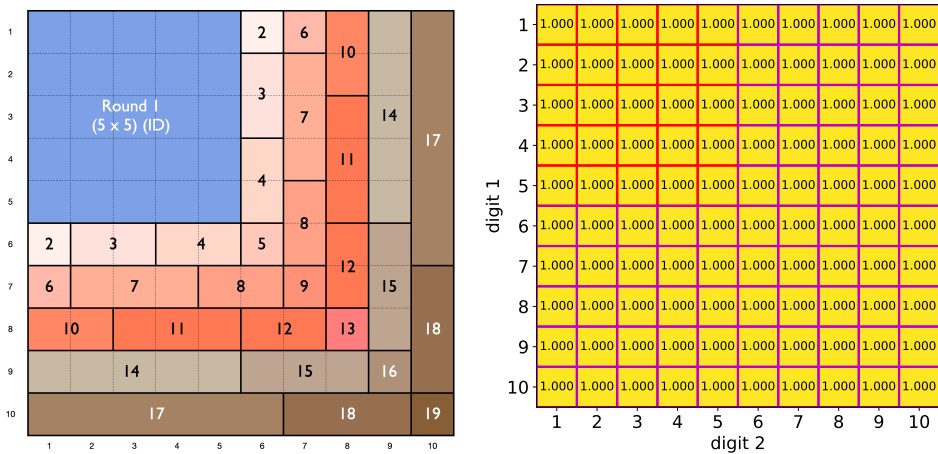

*Figure 23.* Accelerated self-improvement in multiplication. (Left) Accelerated schedule for multiplication. The rows and columns represent the number of digits in the two operands of the multiplication task. The number within each cell indicates the self-improvement round in which the corresponding digit pair is included for training. (Right) Results at round 19. Controlled scheduling progressively incorporates more digit pairs in each round, accelerating the self-improvement process.

## B.5. Results on Pretrained Models

Figure 24 shows the self-improvement results for LoRA finetuning Llama-1B and Llama-3B on the reverse addition task. Pretrained models show more extrapolation than from-scratch models.

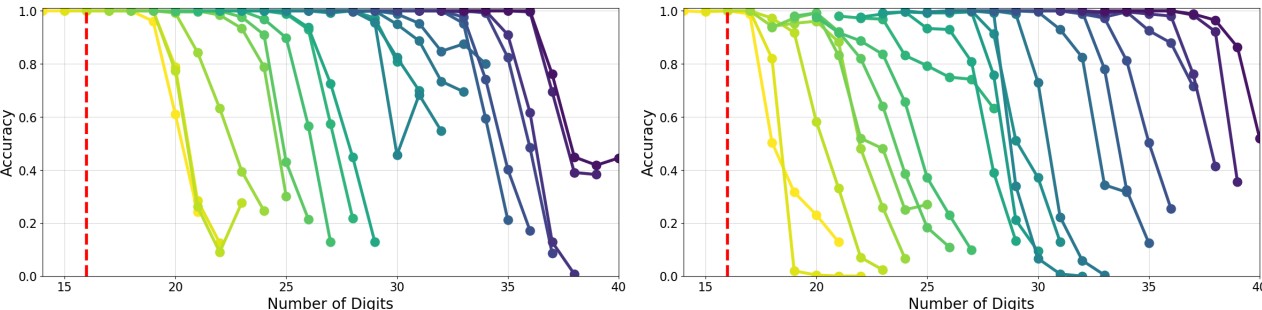

*Figure 24.* Reverse addition results for pretrained models. (Left) Llama-1B model. (Right) Llama-3B model. Larger models exhibit better extrapolation performance across rounds of self-improvement.

## B.6. Additional Error Analysis on Reverse Addition

### B.6.1. PATTERNS IN MODEL MISTAKES

We can categorize all mistakes into two bins. At each digit position, either the model drop the digit, or that it outputs a wrong digit. Since these two kinds of mistakes are entangled in practice, we use a string matching algorithm to compare the model output and predictions and obtain the best guess. In figure 25, we find that digit drops by the model are concentrated near the send of the sequence, and wrong digits are most often off by 1.

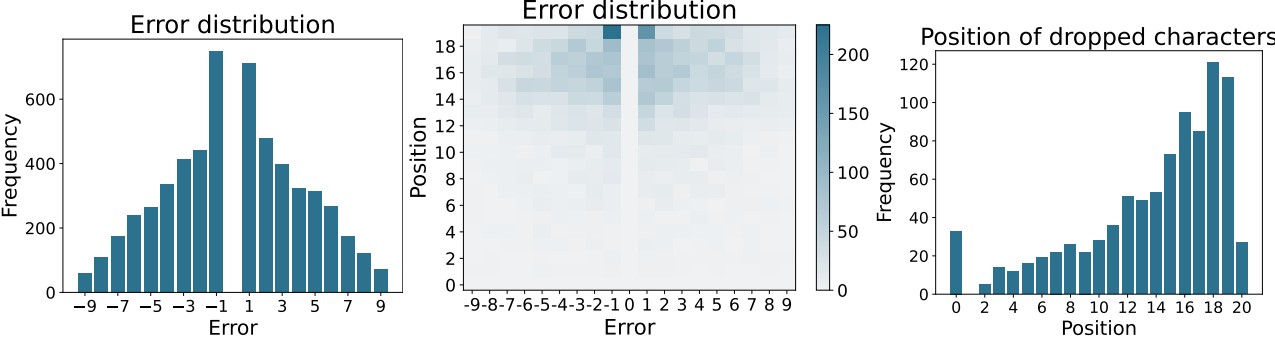

*Figure 25.* Patterns in model errors. (Left) Most incorrect digits are off by 1. (Middle) Errors cluster near the end of the sequence. (Right) Digit drop errors are strongly location-dependent.

Additionally, Figure 26 shows that when models generate incorrect answers, the first mismatch with the ground truth typically occurs near the final digits of the sequence (i.e., near the most significant digit in reverse addition). These observations inform our systematic error simulations, which are used to analyze the error avalanche phenomenon in Section 8.

## B.7. Additional Experiments on Label Noise and Robustness

**Models can Generalize Despite Memorizing Past Mistakes**    Since self-improvement involves recycling model predictions into training data, an important question is whether the model continues making mistakes on previously incorrect examples. To investigate this, we isolate incorrect self-generated samples and evaluate the model's performance on them. As shown in Figure 27, the model struggles to rectify these errors. Accuracy on incorrect training examples decreases over successive rounds, suggesting that repeated exposure to errors reinforces them rather than correcting them.

However, memorizing past mistakes does not necessarily cause an error avalanche. The model under self-improvement often generalize to higher difficulties while treating the incorrect samples as outliers. For example, Figure 13 shows that after 20

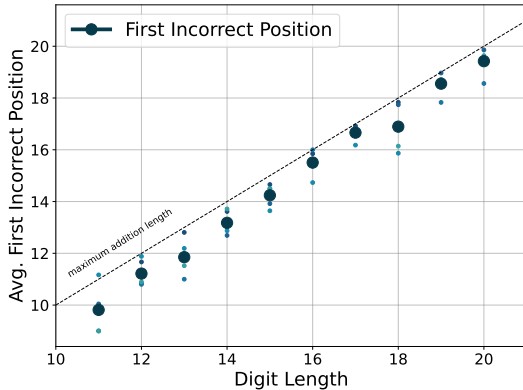

*Figure 26.* he first incorrect digit in model outputs tends to occur near the most significant digit in reverse addition.

rounds of self improvement, the model can tolerate a surprisingly large amount of label noise, from both uniform noise and structured noise. This suggests that while individual mistakes persist, they do not necessarily hinder overall generalization.

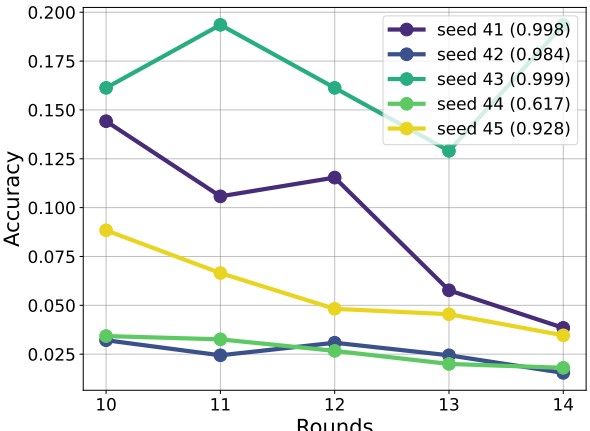

*Figure 27.* Models memorize their mistakes. Accuracy on incorrect training examples (of $\mathcal{D}_9$) decreases with additional self-improvement rounds, indicating that repeated exposure reinforces memorization of errors instead of correcting them.

**Robustness against Random Labels**    To further examine the model's resilience to errors in data, we introduce randomization into the labels during training. Correct labels are replaced with random numbers of the *same length* with probabilities 1, 0.8, 0.5, 0.2, 0.1, and 0. A probability of 1 corresponds to entirely incorrect labels, while 0 indicates fully correct data.

The model is initially trained on 1-10 digit reverse addition and further trained across 8 self-improvement rounds, using self-generated data of lengths 11-18 digits. We then construct a dataset of 19-digit data with randomized labels, denoted as $\mathcal{D}_9^{\text{rand}}$. The model is fine-tuned on a combined dataset consisting of the original dataset $\mathcal{D}_0$, self-improved datasets $\mathcal{D}_1, \ldots, \mathcal{D}_8$, and $\mathcal{D}_9^{\text{rand}}$.

Results in Figure 28 show that the models can tolerate substantial random label noise, maintaining robust performance even when up to 80% of the training data is corrupted. This demonstrates the model's resilience to random errors in the training data and its ability to self-correct such mistakes during learning.

**Model Bias vs. Random Labels.**    Interestingly, biases in self-generated data are more detrimental than uniformly random label noise. As shown in Figure 28, models trained with self-improved data perform worse than random-labeled data of comparable accuracy, given the same dataset size and fine-tuning steps. This suggests that the inherent biases in self-generated data hinder generalization more than randomly introduced noise.

These observations align with findings from Bayat et al. (2024), which highlight that memorization alone does not harm generalization; instead, the combination of spurious correlations undermines learning. Despite memorizing mistakes in

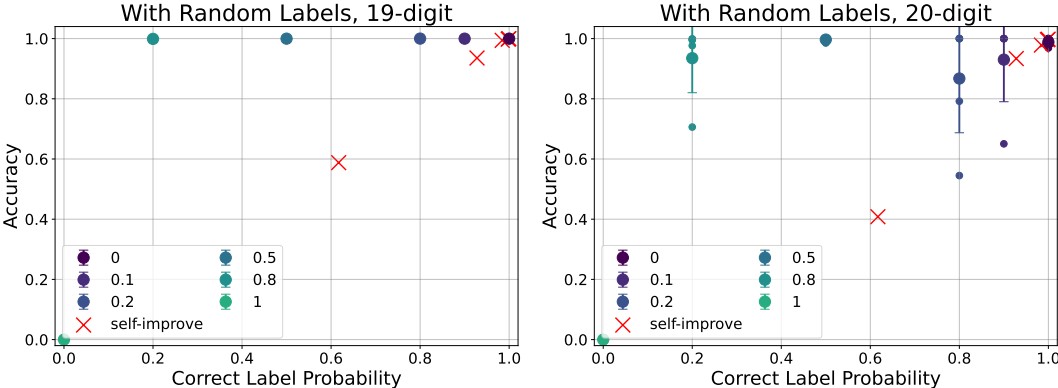

*Figure 28.* Effect of training on randomized labels. The model is trained on 1-10 digit data, further fine-tuned on 11-18 digit self-generated data over 8 self-improvement rounds, and additionally fine-tuned on 19-digit data with varying probabilities of random label replacement. (Left) Accuracy on 19-digit data. (Right) Accuracy on 20-digit data. The results demonstrate that while the model can self-correct random errors, biases from self-improved data can result in worse performance compared to models trained on random-labeled data of similar accuracy.

self-generated data, the model's overall performance at the same difficulty level often exceeds the quality of the training data.

### B.7.1. EFFECT OF SELF-GENERATED DATA QUANTITY ON PERFORMANCE

We investigate how the quantity of self-generated training data impacts model performance. We first train 10 base models $M_0^{(s)}$ ($s = 1, \ldots, 10$) on a supervised 1-10 digit reverse addition dataset $\mathcal{D}_0^s$, each using a different random seed. These models are categorized based on their accuracy on 11-digit addition: low-performing models (less than 98% accuracy) are represented with red colors, while high-performing models (more than 98% accuracy) are depicted with blue colors.

To study the effect of dataset size, we generate self-improvement datasets $\mathcal{D}_1^s = \{(x_i, M_0^{(s)}(x_i))\}_{i=1}^{N_1}$ of varying sizes ($N_1 = 10,000, 50,000, 100,000, 500,000, 1,000,000$). Each model is then trained on the combined dataset $\mathcal{D}_0^s \cup \mathcal{D}_1^s$. The number of incorrect examples in each self-generated dataset is approximately $N_1 \times (1 - 11$-digit accuracy of $M_0)$.

Results in Figure 29 show that for low-performing models, increasing the quantity of self-generated data (which is of lower quality) degrades performance on both in-distribution (11-digit) and out-of-distribution (12-digit) addition. In contrast, for high-performing models, the relationship between the number of self-generated examples and performance is less clear. The total number of 11-digit examples seen during training remains constant across experiments, with smaller datasets being repeated more often. This suggests that exposure to a greater diversity of incorrect examples can bias the model more negatively.

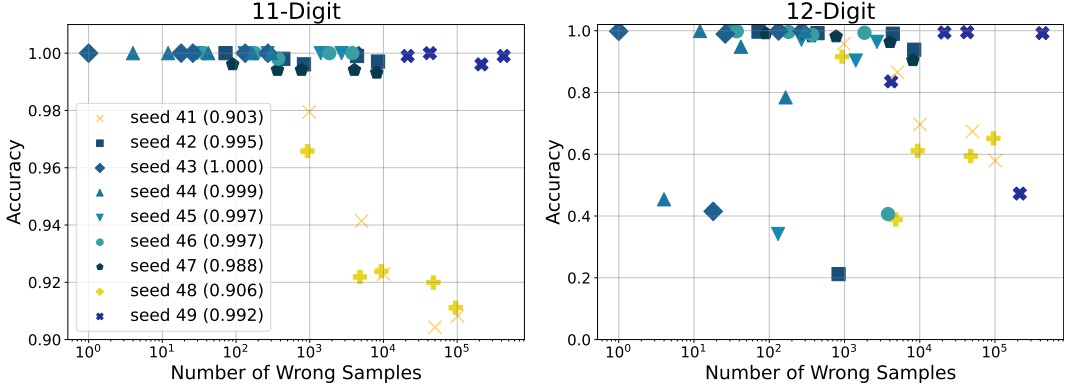

*Figure 29.* Effect of self-generated training data quantity and quality on model performance. Each model is trained on $\mathcal{D}_0$ (1-10 digit addition) and self-generated $\mathcal{D}_1$ (11-digit addition), then evaluated on 11-digit (in-distribution) and 12-digit (out-of-distribution) test performance. For low-performing models, increasing the quantity of self-generated data leads to degraded performance. For high-performing models, the impact of dataset size is less clear.

## C. Experimental Setup

### C.1. Model

For all experiments, we use a Decoder-only Transformer architecture. Specifically, for all experiments except for pretrained models settings, we use the Llama architecture (AI@Meta, 2024), except we remove the rotary positional encoding. For the inputs format, we have one example per line, and stack all example on the batch dimension. Since the examples can have variable length, we pad each line on the right to the maximum length in the batch. We exclusively use a character level tokenizer. For pretrained models, we replace the default tokenizer with our character tokenizer, while keeping the embedding component of the pretrained model unchanged.

*Table 3.* Model Parameters

| Model | Self-Attn Layers | Num Heads | Embedding Dim |
|---|---|---|---|
| From-Scratch | 6 | 6 | 384 |
| Llama 3 1B | 24 | 16 | 1024 |
| Llama 3 3B | 32 | 32 | 2048 |

### C.2. Data Formats and Data Sampling

#### C.2.1. DATA GENERATION AND SAMPLING

We generate an initial supervised training dataset $\mathcal{D}_0$ of up to a fixed difficulty level $d_0$ by uniformly sampling the difficulty level $d \leq d_0$, followed by independent sampling of the data conditioned on the difficulty. Denoting the input as $x_i$, labels as $y_i$,

$$\mathcal{D}_0 = \{(x_i, y_i)\}_{i=1}^{N_0}, \quad \text{where Difficulty}(x_i) \leq d_0.$$

For arithmetic tasks such as addition or multiplication, each problem instance is represented as a tuple $x_i = (a_i, b_i)$, with $\mathcal{D}_0$ containing problems of up to $d_0$-digit numbers. The digit lengths $(d_{a_i}, d_{b_i})$ are uniformly sampled from $\{1, \ldots, d_0\}^2$, and the numbers $a_i$ and $b_i$ are uniformly sampled from the ranges $[10^{d_{a_i}-1}, 10^{d_{a_i}} - 1]$ and $[10^{d_{b_i}-1}, 10^{d_{b_i}} - 1]$, respectively.

For string manipulation tasks (e.g., copying or reversing), we uniformly sample string lengths up to $d_0$ and generate random sequences. Similarly, for maze-solving tasks, we uniformly sample the number of hops or total nodes in the maze and generate random graphs that satisfy these constraints. This strategy ensures balanced coverage across all difficulty levels up to $d_0$.

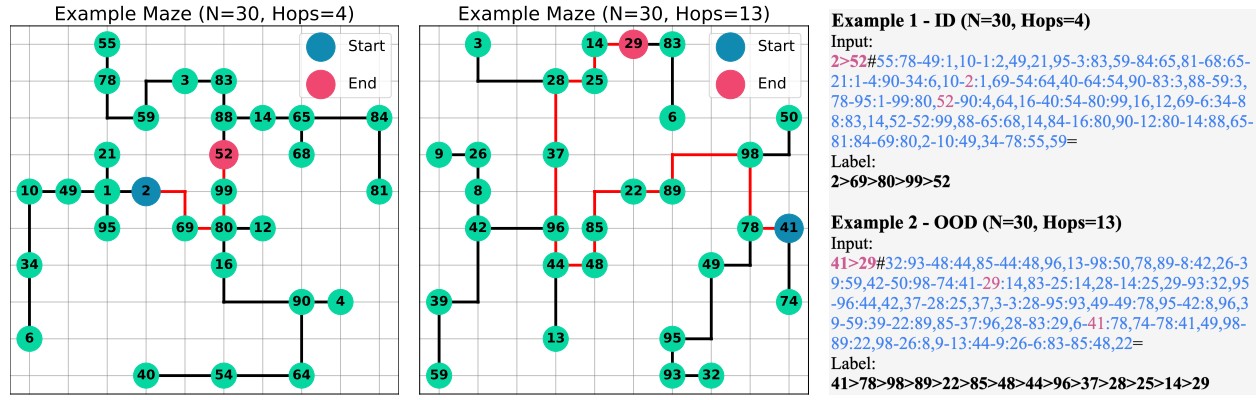

*Figure 30.* Maze-solving task with $N = 30$ nodes. (Left & Middle) Visualization of the maze task with 4 hops (ID) and 13 hops (OOD). (Right) Example of the data format: the input specifies the start and end nodes along with the graph structure, and the output lists the shortest path as hops. The labeled training dataset includes paths of up to 9 hops, with difficulty increased by adding one hop in each subsequent round.

### C.2.2. MULTIPLICATION

We adopt a data format similar to Deng et al. (2024), where the input prompt is `9172*9431=`, and the label expands the multiplication into steps, such as: `17442+067801(132331)+0075180(1398490)+00091720=13976630`. Each step includes the intermediate results (in parentheses) representing partial products formed by multiplying the first operand with each digit of the second operand.

The data format is inherently asymmetrical. For example, an $m$-by-$n$ multiplication requires $n$ intermediate steps, where each step corresponds to multiplying the $m$-digit number by one digit of the $n$-digit number. Conversely, an $n$-by-$m$ multiplication involves $m$ intermediate steps of multiplying the $n$-digit number by each digit of the $m$-digit number.

### C.2.3. MAZE

---

**Listing 1** Code for the maze format generation used

```python
def create_tree_with_hops_wilson(total_nodes, num_hops):
    import networkx as nx

    # Step 1: Create the main path with num_hops
    graph = nx.path_graph(num_hops + 1)

    # Step 2: Add extra nodes to the tree with random walk
    current_nodes = list(graph.nodes())
    new_nodes = list(range(num_hops + 1, total_nodes))

    while new_nodes:
        new_node = new_nodes.pop()
        # random walk to reach graph
        walk = [new_node]
        while walk[-1] not in current_nodes:
            # choose random node from current & new nodes
            random_node = random.choice(current_nodes + new_nodes)
            walk.append(random_node)
            if random_node in new_nodes:
                new_nodes.remove(random_node)
        # add edges
        for i in range(len(walk) - 1):
            graph.add_edge(walk[i], walk[i + 1])
        current_nodes.append(new_node)

    # Step 3: Set the start and end nodes for the main path
    start_node = 0
    end_node = num_hops

    return graph, start_node, end_node

def format_graph(graph, start_node, end_node):
    # Assign random labels to nodes
    node_labels = assign_labels(graph.nodes(), label_range=(1, 99))

    # Get the shortest path (in terms of edge count) from start_node to end_node
    shortest_path = nx.shortest_path(graph, source=start_node, target=end_node)

    # Format the path as a string
    path_labels = [node_labels[node] for node in shortest_path]
    path_string = ">".join(map(str, path_labels))

    # Format start and end nodes
    start_label = node_labels[start_node]
    end_label = node_labels[end_node]
    start_end_str = f"{start_label}>{end_label}#"

    # Build graph_str with end_node connections at the end
    graph_str = ""
    start_node_str = ""   # Temporary storage for the start_node part
    end_node_str = ""   # Temporary storage for the end_node part

    # randomize the order of nodes
    random_nodes = list(graph.nodes())
    random.shuffle(random_nodes)
    for node in random_nodes:
        node_label = node_labels[node]
        # randomize the order of neighbors
        random_neighbors = list(graph.adj[node])
        random.shuffle(random_neighbors)
        neighbor_labels = [node_labels[neighbor] for neighbor in random_neighbors]
        graph_str += f"{node_label}:" + ",".join(map(str, neighbor_labels)) + "-"

    # Combine everything, placing the end_node last
    graph_str = start_node_str + graph_str + end_node_str

    return start_end_str + graph_str[:-1] + "=", path_string, node_labels
```

### C.3. Experimental Settings

#### C.3.1. HYPERPARAMETER CONFIGURATIONS

In this section, we provide a detailed overview of the hyperparameter configuration used in our experiments in Table 4 and 5. To enhance memory efficiency and training speed, we employ flash attention and tf32, bfloat16. Our experiments are run using PyTorch 2.4 and CUDA 12.1. Detailed dependencies are provided in our github repository[2]. We use Warmup stable decay (Wen et al., 2024) as the learning rate schedule. In table 4 and 5, the number of constant LR steps is equal to the total training steps minus the sum of warmup and decay steps.

Table 4 shows the training hyperparameters for the initial training phase on labeled data $D_0$. Table 5 shows the hyperparameters for each the self-improve training rounds on $D_{1,...,R}$.

*Table 4.* Hyperparameters for initial training on labeled data

| Task | Batch Size | Optimizer | LR | Betas | Epsilon | Iterations | Warmup Iter | Decay Iter | Wt decay |
|---|---|---|---|---|---|---|---|---|---|
| Reverse Addition | 1024 | AdamW | 5e-4 | $(0.9, 0.99)$ | 1e-12 | 10000 | 1000 | 2000 | 0.1 |
| Reverse Addition (Llama 3 3B) | 128 | AdamW | 1e-4 | $(0.9, 0.99)$ | 1e-12 | 1200 | 120 | 600 | 0.1 |
| Reverse Addition (Llama 3 1B) | 128 | AdamW | 1e-4 | $(0.9, 0.99)$ | 1e-12 | 1200 | 120 | 600 | 0.1 |
| Copy/Reverse | 1024 | AdamW | 5e-4 | $(0.9, 0.99)$ | 1e-12 | 5000 | 500 | 1000 | 0.1 |
| Forward Addition | 1024 | AdamW | 5e-4 | $(0.9, 0.99)$ | 1e-12 | 10000 | 1000 | 1000 | 0.1 |
| Multiplication | 1024 | AdamW | 5e-5 | $(0.9, 0.99)$ | 1e-12 | 10000 | 1000 | 2000 | 0.1 |
| Maze (hops) | 1024 | AdamW | 5e-4 | $(0.9, 0.99)$ | 1e-12 | 25000 | 2500 | 3500 | 0.1 |
| Maze (nodes) | 512 | AdamW | 5e-4 | $(0.9, 0.99)$ | 1e-12 | 12000 | 1200 | 2800 | 0.1 |

*Table 5.* Hyperparameters for self-improvement rounds

| Input Format | Batch Size | Optimizer | LR | Betas | Epsilon | Iterations | Warmup Iter | Decay Iter | Wt decay |
|---|---|---|---|---|---|---|---|---|---|
| Reverse Addition | 1024 | AdamW | 5e-4 | $(0.9, 0.99)$ | 1e-12 | 1500 | 0 | 1500 | 0.1 |
| Reverse Addition (Llama 3 3B) | 128 | AdamW | 1e-4 | $(0.9, 0.99)$ | 1e-12 | 600 | 0 | 600 | 0.1 |
| Reverse Addition (Llama 3 1B) | 128 | AdamW | 1e-4 | $(0.9, 0.99)$ | 1e-12 | 600 | 0 | 600 | 0.1 |
| Copy/Reverse | 1024 | AdamW | 5e-4 | $(0.9, 0.99)$ | 1e-12 | 500 | 0 | 500 | 0.1 |
| Forward Addition | 1024 | AdamW | 5e-4 | $(0.9, 0.99)$ | 1e-12 | 3000 | 0 | 1000 | 0.1 |
| Multiplication | 1024 | AdamW | 5e-5 | $(0.9, 0.99)$ | 1e-12 | 3000 | 0 | 1000 | 0.1 |
| Maze (hops) | 1024 | AdamW | 2e-4 | $(0.9, 0.99)$ | 1e-12 | 5000 | 500 | 1000 | 0.1 |
| Maze (nodes) | 1024 | AdamW | 2e-4 | $(0.9, 0.99)$ | 1e-12 | 4000 | 400 | 1000 | 0.1 |

#### C.3.2. SELF-IMPROVEMENT SETTING FOR EACH TASK

**Reverse Addition.** The initial supervised dataset $\mathcal{D}_0$ contains 2 million examples of reverse addition, with operand lengths ranging from 1 to 16 digits. This dataset is used to train the model for 10,000 steps. In subsequent self-improvement rounds, we sample 50,000 additional training examples at each round, extending the operand length by one digit. Specifically, at self-improvement round $r$, the self-generated data $\mathcal{D}_r$ consists of length-$16 + r$ examples produced by the model $M_r$. The model is fine-tuned on the combined dataset $\mathcal{D}_0 \cup \mathcal{D}_1 \cup \cdots \cup \mathcal{D}_r$ for 1,500 steps, resulting in an improved model $M_{r+1}$.

**String Copy & String Reverse.** The initial training set $\mathcal{D}_\prime$ consists of 2 million examples of strings of length 1 to 10. The vocabulary of the string is the digits 0 to 9. For each subsequent round $r$, we sample $D_r$ consisting of $50,000$ examples of length $10 + r$ from the model $M_r$. Then we continue training $M_r$ on the combined dataset $D_1 \cup \cdots \cup D_r$ for 500 steps to obtain $M_{r+1}$.

**Forward Addition** The models are initially trained on a dataset $\mathcal{D}_0$ containing 2 million labeled examples of forward addition, with operand lengths ranging from 1 to 10 digits. This initial training phase spans 10,000 steps. In each subsequent self-improvement round, we generate 50,000 additional training examples, incrementally extending the operand length by one digit. Specifically, at self-improvement round $r$, the self-generated dataset $\mathcal{D}_r$ contains length-$10 + r$ examples produced by the model $M_r$. The model is then fine-tuned for 3,000 steps on the combined dataset $\mathcal{D}_0 \cup \mathcal{D}_1 \cup \cdots \cup \mathcal{D}_r$, resulting in an updated model $M_{r+1}$.

---

[2]https://github.com/JackCai1206/arithmetic-self-improve

**Multiplication**    The model is initially trained on 5 million $n$-by-$n$ multiplication examples with $n = 5$. Directly introducing $n + 1$-by-$n + 1$ examples results in poor performance, hence, we adopt a more fine-grained difficulty schedule. In each self-improvement round, we incrementally increase one operand by one digit, sampling $n + 1$-by-$m$ and $m$-by-$n + 1$ examples, where $m$ grows from 1 to $n + 1$. This gradual progression allows the model to adapt incrementally to larger operand sizes, making the transition to harder examples more manageable.

For data filtering, we use the following setting: for length filtering, we remove self-generated samples where the output length is shorter than the longest output in the batch by more than 10 tokens. This helps eliminate incorrect solutions that omit intermediate steps. For majority voting, we train five models in parallel using different random seeds and retain only those data points where at least 4 out of the 5 models produce the same output. This strategy ensures that only high-consensus, reliable data points are used for training.

**Maze Solving - Increasing Hops.**    The model is first trained on a dataset $\mathcal{D}_0$ containing 5 million labeled maze-solving examples, where the number of nodes is fixed at $N = 30$ and paths range from $h = 1$ to $h = 9$ hops. This initial training phase spans 25,000 steps. In subsequent self-improvement rounds, we generate 50,000 additional training examples, increasing $h$ by 1, and fine-tune the model for 5,000 steps per round. We experiment with both unfiltered training data and majority voting, where only outputs agreed upon by all 3 models are retained.

**Maze Solving - Increasing Nodes.**    The model is first trained on a dataset $\mathcal{D}_0$ containing 5 million labeled maze-solving examples, with a fixed hop count $h = 9$ and node counts ranging from $N = 10$ to $N = 30$. This initial training lasts 12,000 steps. In self-improvement rounds, the number of nodes $N$ is increased by 3 per round, generating 50,000 additional training examples at each step and fine-tuning for 4,000 steps. We compare training without filtering against majority voting, where only outputs agreed upon by all 3 models are kept.

**Ablation Task - Pretrained Models**    To maintain consistency in tokenization, we use character-level tokenization instead of the default tokenizer of the Llama models. We use LoRA (Hu et al., 2021) with $r = 64$ and $\alpha = 128$ for Llama-1B, and $r = 32$ and $\alpha = 128$ for Llama-3B. In the initial round, we train for 1200 steps with a learning rate schedule that includes 10% warm-up steps to a constant learning rate of 1e-4, followed by 20% cosine decay steps to a final learning rate of 1e-6. For subsequent rounds, we train for 600 steps per round using a cosine decay learning rate schedule without warm-up, starting at 1e-4 and decaying to 1e-6.

# D. Full Results

### D.0.1. RESULTS ON STRING COPY & STRING REVERSE

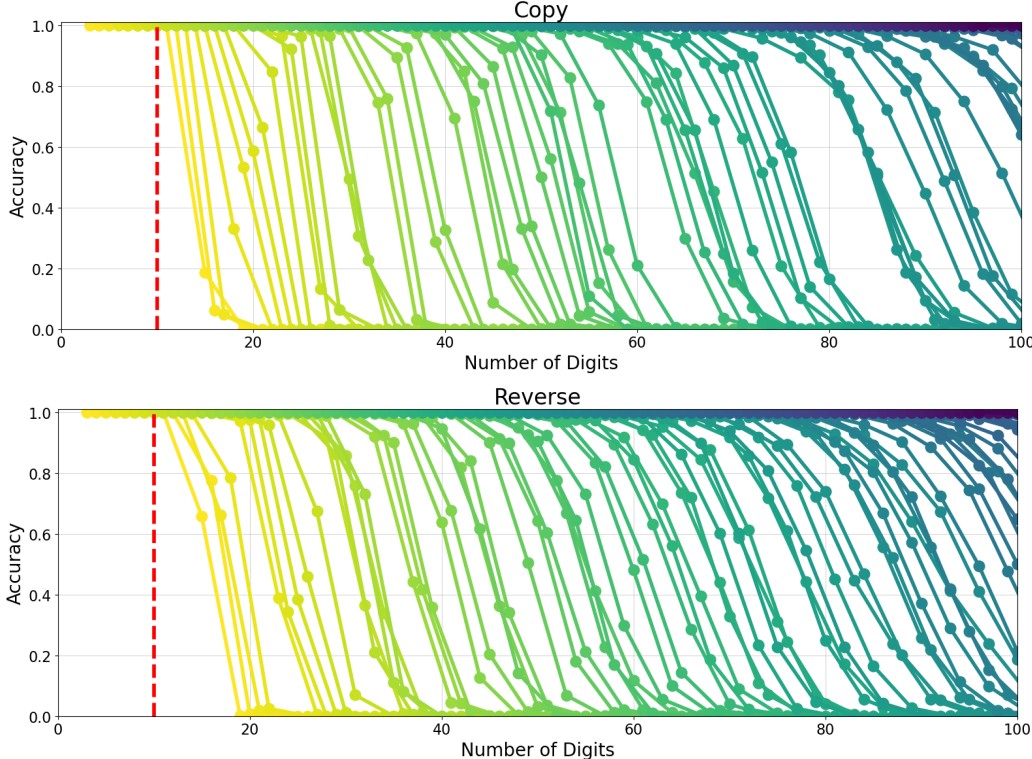

*Figure 31.* Results on string manipulation tasks. (Top) Copying task. (Bottom) Reversing task. The model, initially trained on strings of length 1 to 10, generalizes to strings of over 120 digits through self-improvement.

### D.0.2. RESULTS ON MULTIPLICATION

Each figure represents the average over 5 different models.

### D.0.3. RESULTS ON MAZES

We provide additional evaluation on mazes, based on the validity of moves and correctness of end nodes.

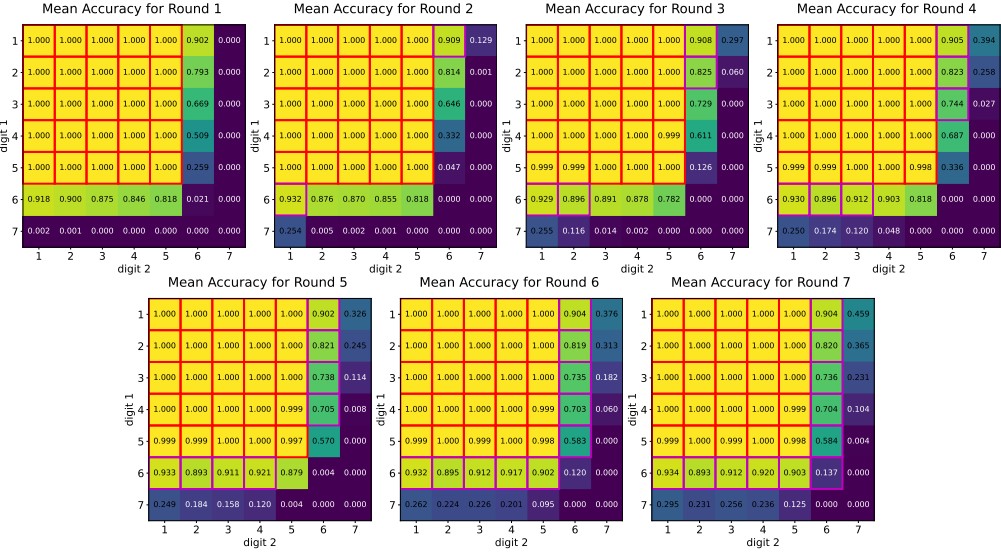

*Figure 32.* Results for multiplication without filtering. Each cell represents the accuracy on $n$-digit by $m$-digit multiplication. Red boxes indicate labeled in-distribution examples, while magenta boxes indicate evaluations after training on self-improved data. The model is initially trained on up to 5-by-5 multiplication. Generalizing to larger multiplications is challenging without data filtering.

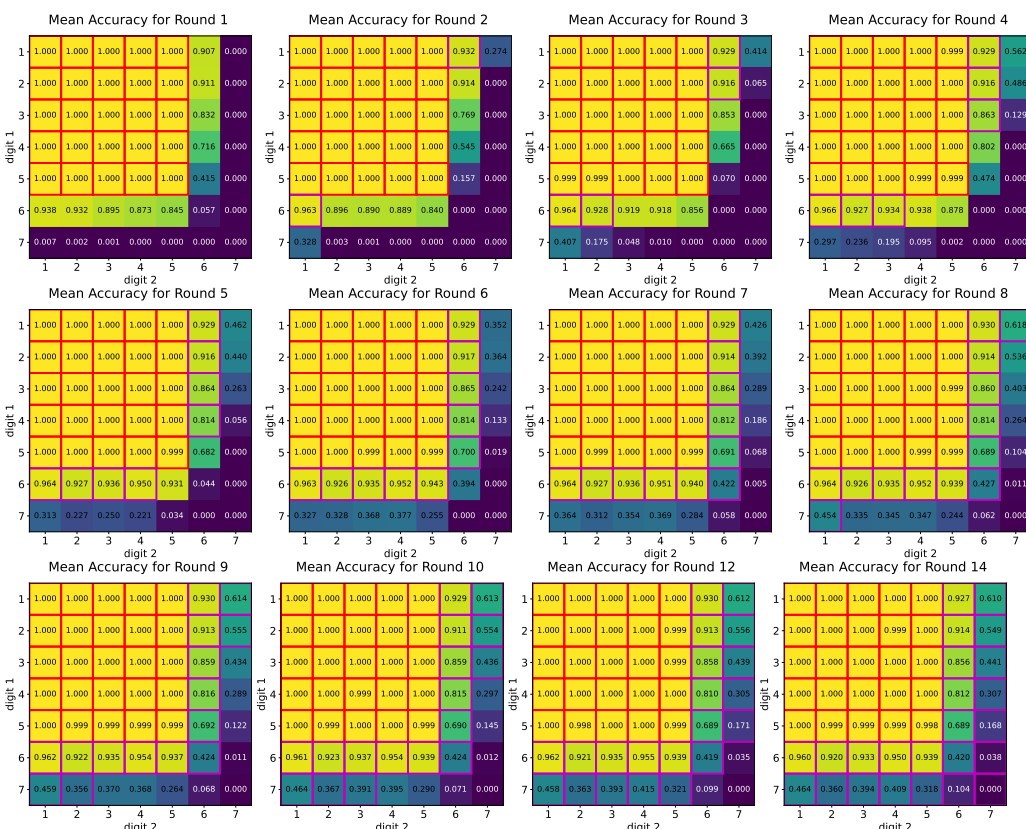

*Figure 33.* Results for multiplication with length filtering with length threshold of 10.

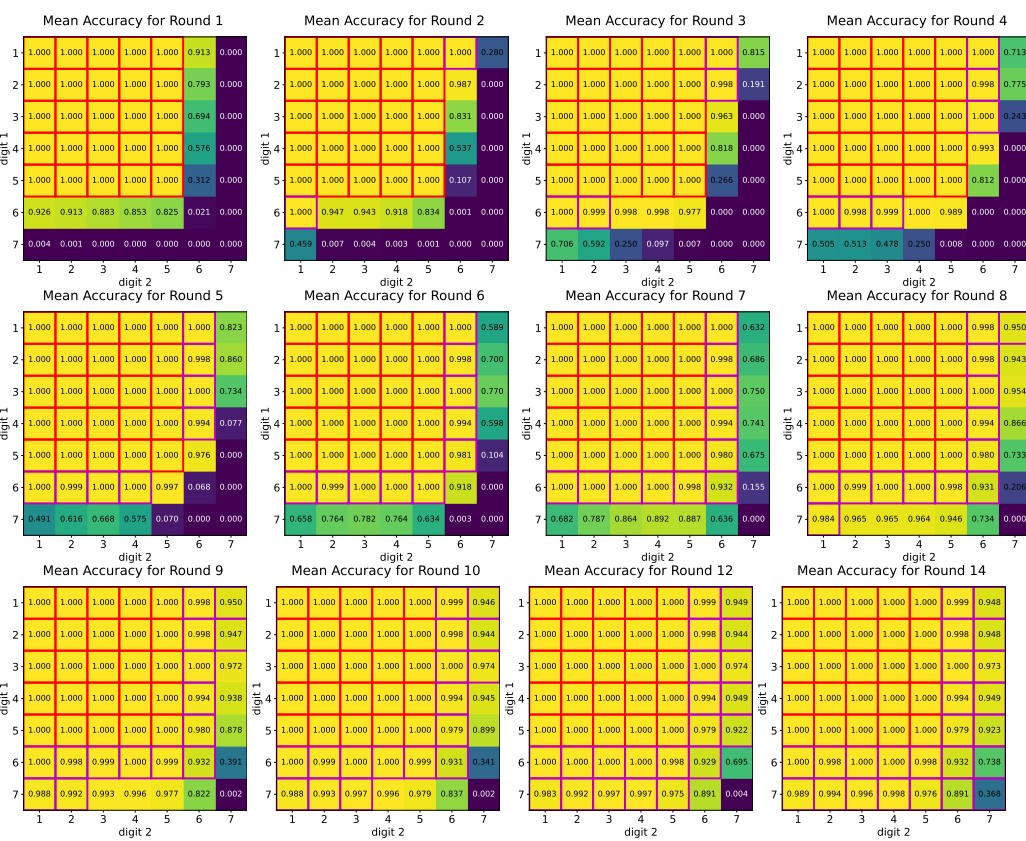

*Figure 34.* Multiplication with majority voting where filtering is based on agreement of at least 4 out of 5 models. Applying majority voting enables effective generalization from $n$-by-$n$ to $(n + 1)$-by-$(n + 1)$ multiplication tasks.

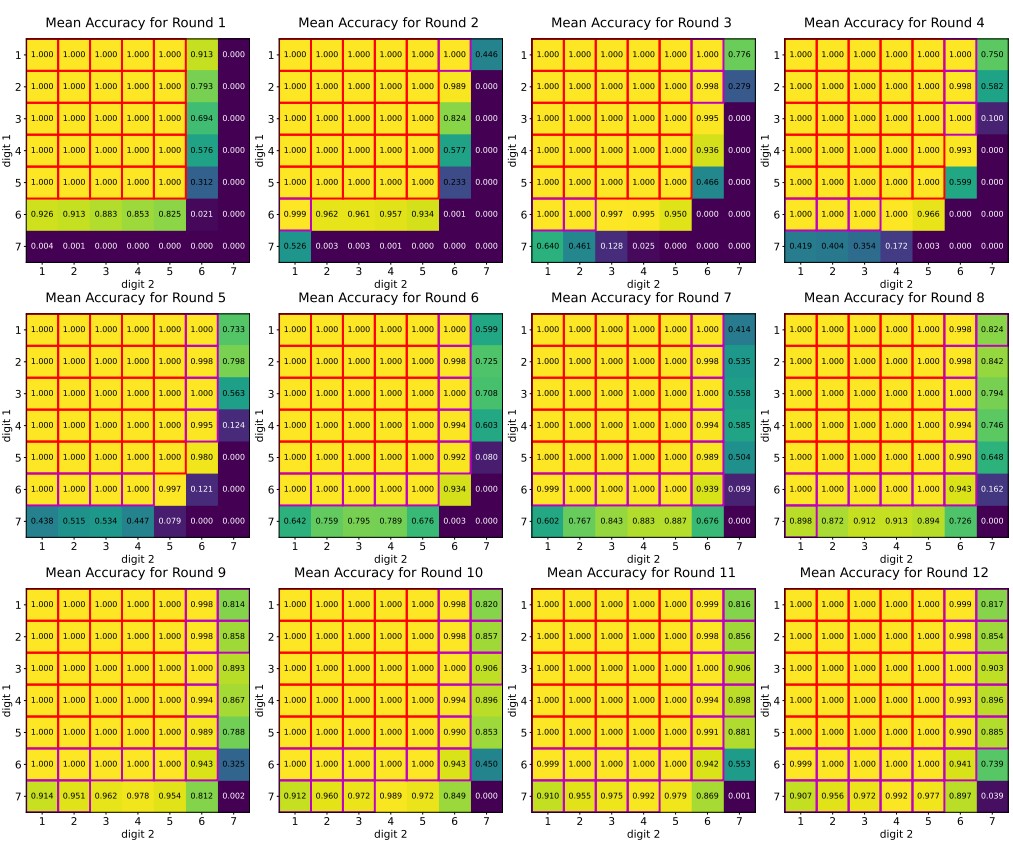

*Figure 35.* Multiplication task with majority voting with shared self-improve data (See Section B.2.4).

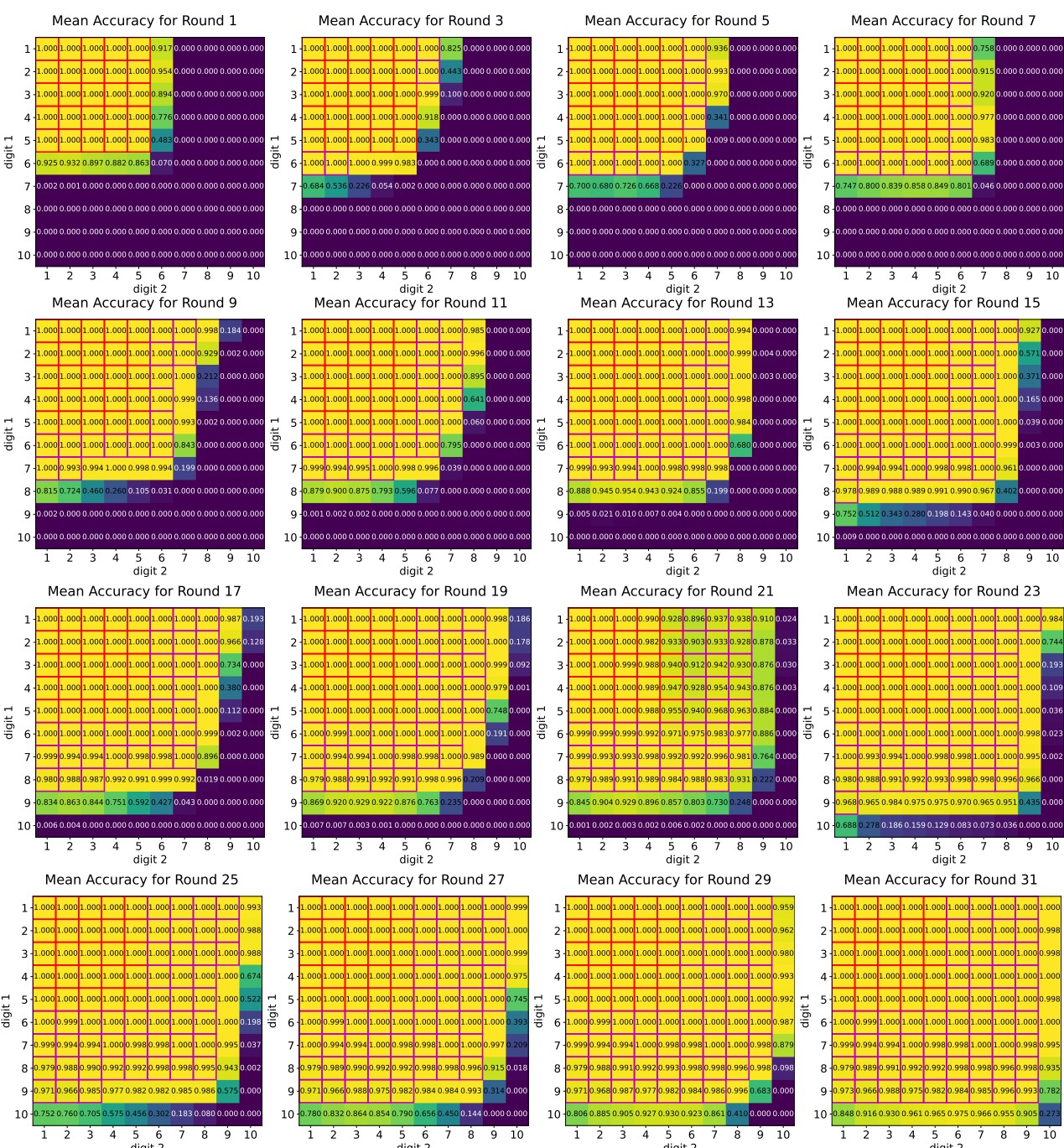

*Figure 36.* Combining majority voting with length filtering. This approach achieves near-perfect length generalization up to $9 \times 9$, and potentially achieving further generalization.

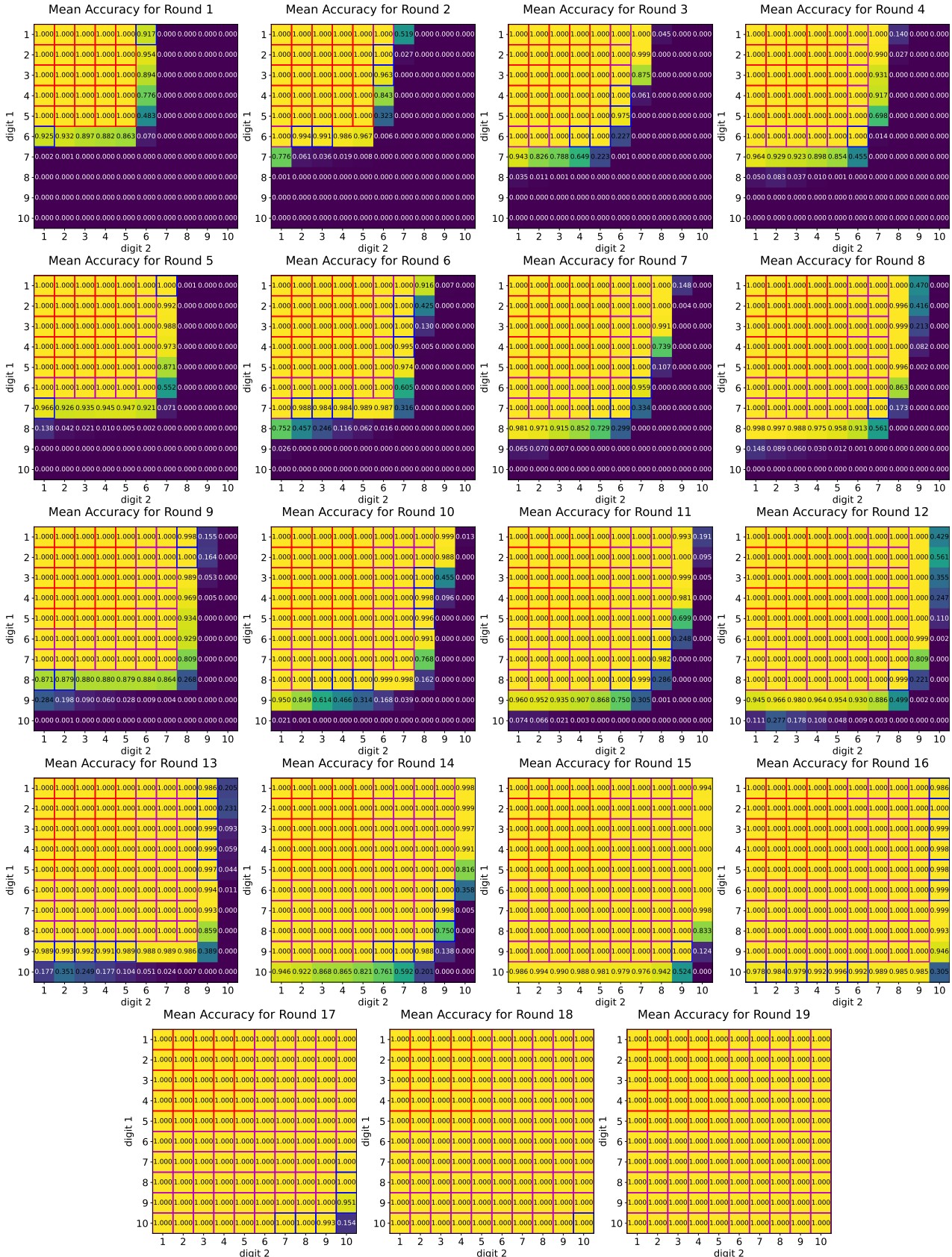

*Figure 37.* Accelerated multiplication. We can significantly reduce the self-improvement rounds by carefully sampling a wider range of difficulties at every round. Perfect length generalization is achieved up to 10-by-10 multiplication with 19 self-improvement rounds.

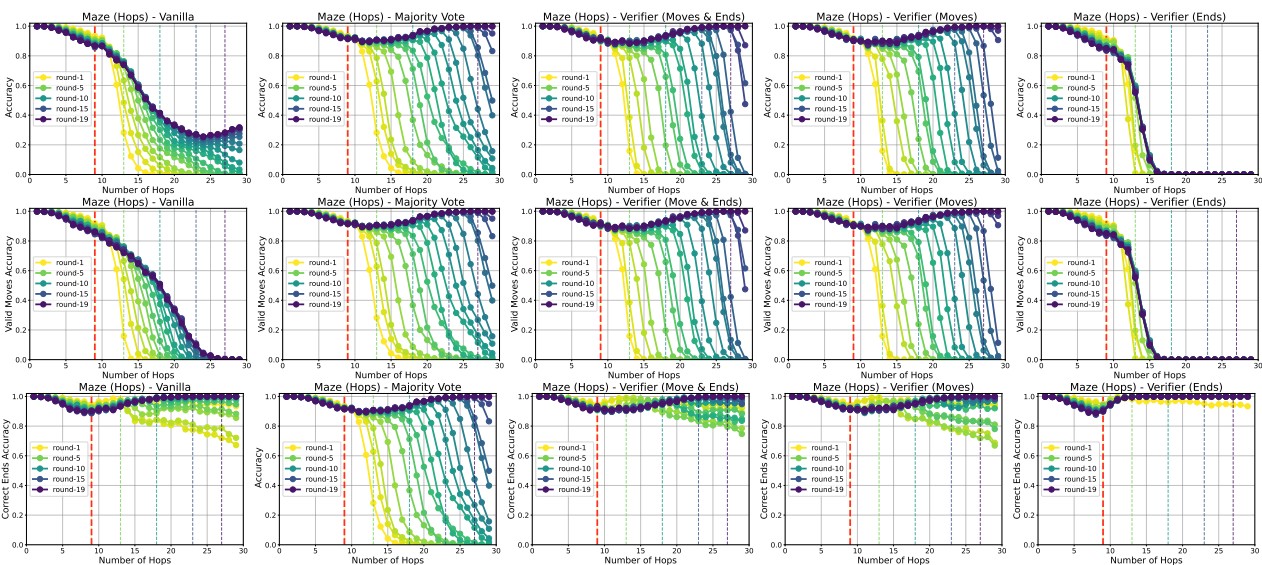

*Figure 38.* Maze solving task with increasing hops. (Top to bottom) Exact match accuracy, move validation accuracy, and end validation accuracy. (Left to right) No data filtering, majority voting based filtering, verifier on both moves and ends, verifier on moves only, verifier on ends only.

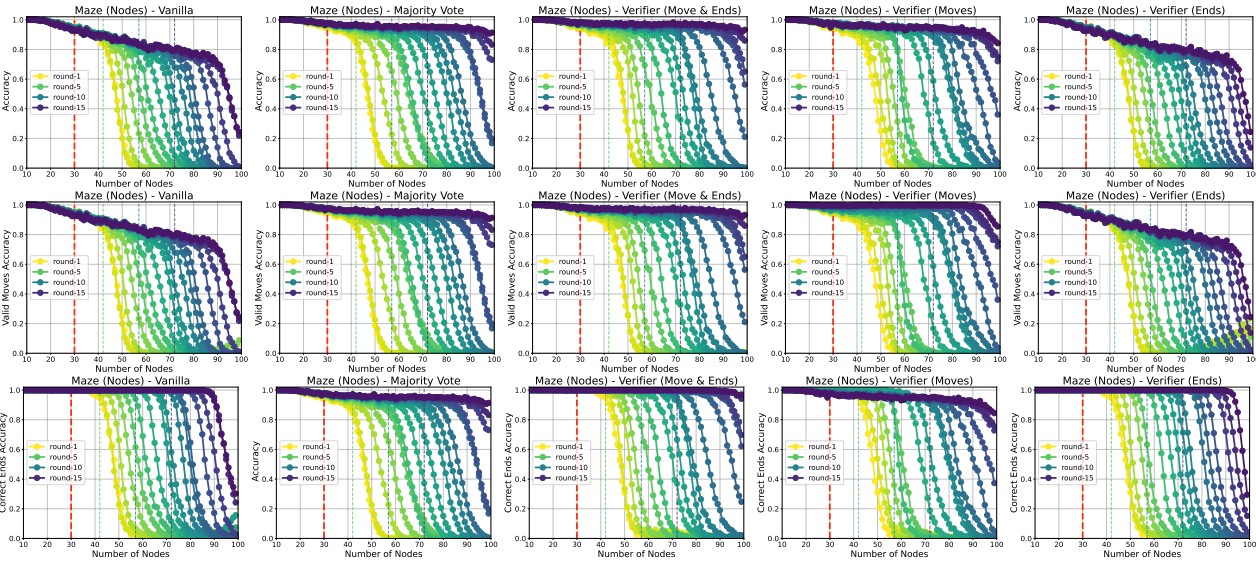

*Figure 39.* Maze solving task with increasing nodes. (Top to bottom) Exact match accuracy, move validation accuracy, and end validation accuracy. (Left to right) No data filtering, majority voting based filtering, verifier on both moves and ends, verifier on moves only, verifier on ends only.

