# OpenReview forum: "Self-Improving Transformers Overcome Easy-to-Hard and Length Generalization Challenges"
_ICML.cc/2025/Conference — ICML 2025 poster_

### Official Review · Reviewer_FEuQ · 2025-03-10

**Overall Recommendation:** 4

**Summary:**

The paper demonstrates the capability of language models for self-improvement: achieving improved performance on problem instances beyond the training data, using model generations alone (without additional labeled data). The work focuses on two settings: length generalization, where the model is trained on shorter sequences and extrapolates to longer ones; and easy-to-hard generalization, where model is trained on easy/simple problem instances and is trained to generalize to harder instances. The authors study different arithmetic problems (addition and multiplication), copying and maze-solving, and show impressive results in all settings. To achieve self-improvement, the authors initially train a model on short/simple instances with fully supervised labels, and then iteratively increase the length/complexity of the problems, training the model on its own generations instead of using labeled data. The authors utilize data filtering techniques -  majority voting and filtering based on length, and demonstrate their effectiveness in achieving out-of-distribution generalization in harder settings. Finally, the authors analyze how errors accumulate during the execution of the self-improvement procedure, and how data filtering can mitigate this.

**Claims And Evidence:**

The method and the results shown in the paper are novel, with very impressive performance on hard tasks like multiplication and maze solving. The paper is very well-written, the ideas are clearly introduced and the authors did a very good job in presentation of the results. The study of error accumulation is also interesting in its own right, and overall the experiments demonstrate the success and limitation of different methods very thoroughly.
Some concerns regarding the results:
- I think that the comparison to other length generalization works is a little misleading. Length generalization is interesting for two reasons: First, because we don't have training data with long sequences; and second, because training on long sequences is very costly. The method presented in the paper addresses the first problem, i.e. - gives a solution that could be applied in cases where we don't have training data with long sequences (although, the method does require input "questions" for longer sequences, also discussed in the limitations section). However, it does not improve the computational cost of training on long sequences, and if anything makes the training cost much higher. While I believe that addressing only the first issue is important, the authors should discuss this limitation explicitly. Additionally, I think that discussing the overall computational cost of the method (and ideally compare it to other methods for length generalization) would be helpful.
- Related to the above, I think the authors should directly compare their results, at least in some of the experiments, to other length generalization methods on the same tasks. The authors claim that other length generalization methods change the positional encoding, architecture and data to address length generalization, in contrast to this paper. However, the paper also uses a somewhat non-standard positional encoding (NoPE). So, how do the results in the paper compare to length generalization methods that make "reasonable" changes (i.e., changing the positional encoding but not the data)?
- In the majortiy voting based filtering, the authors use an ensemble of models, all trained from scratch with different seeds and datasets. This seems much more expensive than sampling with temperature. Does temperature sampling simply not work, requiring this more complex method? I think that discussing the computational cost of this ensemble method would be useful, and also maybe comment on how it could be improved (e.g., how large should k be for the method to work)?

More minor comments:
- In the addition experiments, the authors claim that they achieve generalization to over 100 digits, but from the plots it seems that performance could keep improving. How far does the generalization actually reach? Did the authors run experiments beyond this range?
- The description of relative length filtering is unclear. How is $\tau$ used? Are the lengths filtered based on a fixed constant, or relative to the maximal length L?

To summarize, I think this is a very good paper, demonstrating the effectiveness of self-improvement method for achieving out-of-distribution generalization in different settings. The experiments are very convincing, but I believe that proper discussion of computational cost compared to other methods could improve the paper.

**Essential References Not Discussed:**

No.

**Experimental Designs Or Analyses:**

See above.

**Methods And Evaluation Criteria:**

See above.

**Other Comments Or Suggestions:**

N/A

**Other Strengths And Weaknesses:**

N/A

**Questions For Authors:**

See above.

**Relation To Broader Scientific Literature:**

See above.

**Theoretical Claims:**

N/A

---

> ### Author Rebuttal · Authors · 2025-04-01
>
> Thank you for your thoughtful review and appreciating the novelty, clarity, and thoroughness of our approach and experiments. We address your specific comments and suggestions below:
>
> **Computational Cost**
> > Length generalization is interesting for two reasons: First, because we don't have training data with long sequences; and second, because training on long sequences is very costly. ...  While I believe that addressing only the first issue is important, the authors should discuss this limitation explicitly. Additionally, I think that discussing the overall computational cost of the method (and ideally compare it to other methods for length generalization) would be helpful."
>
> We appreciate your highlighting this point. While our manuscript briefly mentions the inherent differences of our method from other length generalization approaches (lines 82, 623-626), we agree that explicitly mentioning computational costs is crucial. Indeed, our iterative training method does require higher computational resources compared to single-round training strategies, as it involves multiple iterations of model training. We will clearly discuss this computational trade-off and compare it with other methods explicitly in the revised manuscript.
>
> **Comparison with Positional Encoding Method**
> >  I think the authors should directly compare their results, at least in some of the experiments, to other length generalization methods on the same tasks.
>
> We acknowledge your suggestion to explicitly compare our approach with other positional encoding (PE) methods. We want to emphasize that our framework is fundamentally architecture-agnostic and can be combined seamlessly with various positional encodings, including both NoPE and RoPE.
> Specifically, we used NoPE in experiments involving small models trained from scratch due to its better length generalization properties compared to absolute position encoding (APE) or RoPE. Conversely, for our pretrained model experiments, we employed pretrained LLaMA models utilizing RoPE. Hence, the choice of positional encoding is orthogonal to our core contribution. We will clarify and elaborate this point.
>
> **Clarification on Majority Voting Filtering**
> > In the majortiy voting based filtering, the authors use an ensemble of models, all trained from scratch with different seeds and datasets. This seems much more expensive than sampling with temperature. Does temperature sampling simply not work, requiring this more complex method? I think that discussing the computational cost of this ensemble method would be useful, and also maybe comment on how it could be improved (e.g., how large should k be for the method to work)?
>
> We did not consider temperature scaling because our tasks are structured to have a specific input-output format. Thus, temperature sampling did not yield sufficient output diversity. Specifically, we observed very high confidence (over 0.999) in top-1 outputs using beam search (for reverse addition task), indicating minimal diversity even with temperature sampling. However, we anticipate temperature scaling could be more effective for pretrained models that can generate more varied outputs.
>
> Regarding computational cost, we conducted ablation studies for majority voting in Appendix Section B.2.4, exploring data cost and performance trade-offs. Given computational costs scale linearly with the number of ensemble models, we will add explicit commentary on this trade-off, as well as provide new results on how accuracy and performance vary with ensemble size (k) and consensus thresholds in our revision.
>
> **Q1. Generalization beyond 100 digits**
> > In the addition experiments, the authors claim that they achieve generalization to over 100 digits, but from the plots it seems that performance could keep improving. How far does the generalization actually reach? Did the authors run experiments beyond this range?
>
> Our experiments show that self-improvement can continue indefinitely as long as data accuracy remains high. We halted our experiments around 100-digit sequences due to memory constraints, but since performance remained stable, we anticipate continued successful generalization beyond this length.
>
> **Q2. Relative Filtering**
> > The description of relative length filtering is unclear. How is used? Are the lengths filtered based on a fixed constant, or relative to the maximal length L?
>
> We apologize for the unclear description. Currently, relative length filtering uses fixed thresholds (e.g., threshold value of 2 for forward addition and 10 for multiplication). We will clarify this clearly in our revised manuscript. Following the suggestion from Reviewer vrfN, we will also add sensitivity analyses for filtering thresholds. Employing thresholds relative to maximal sequence length L would be a valid (and insightful) approach as well, although it is not used in our experiments.

---

### Official Review · Reviewer_MReD · 2025-03-13

**Overall Recommendation:** 2

**Summary:**

This paper presents a self-improvement approach where standard decoder transformer models iteratively generate and learn from their own predictions. The authors show that this self-improvement approach allows models to achieve extreme length generalization, where the length of a test instance can be up to 5x-6x longer than the length of a training instance, for addition, string copy, multiplication, and shortest path problems. For these problems, length generalization corresponds naturally to easy-to-hard generalization, where longer input instances are also harder to solve. The core of the proposed self-improvement approach is a data filtering procedure. Data filtering ensures that the self-generated labels are accurate enough so that they will not degrade the model's performance when it is fine tuned on its own predictions.

**Claims And Evidence:**

Yes

**Essential References Not Discussed:**

NA

**Experimental Designs Or Analyses:**

I checked all experimental designs and analyses in the main paper, as well as the motivation for data filtering discussed in the appendix. The experiments are well designed and thorough.

**Methods And Evaluation Criteria:**

Yes

**Other Comments Or Suggestions:**

NA

**Other Strengths And Weaknesses:**

While I appreciate the comprehensive empirical evaluation and analysis presented in this paper, I think the scope of this paper is very much limited by the simplicity of the tasks considered by the authors. On these tasks, it is natural that the length of the input corresponds nicely to the difficulty of the problem. However, in many (and more realistic) cases, such a correspondence is not that straightforward. More importantly, as the authors show in the paper, the success of self-improvement hinges on the quality of the model's prediction at every round, which in turn is almost entirely dependent on the data filtering procedure. Without the data filtering procedure, the self-improvement framework will lead to much worse results. It is unclear how to design good data filtering procedure on more complex tasks. Therefore, the implications of the findings presented in this paper to more complex settings is not clear. Besides stating this as a limitation, the authors should at least discuss how the results on simple tasks would be useful for more complex tasks.

**Questions For Authors:**

NA

**Relation To Broader Scientific Literature:**

Achieving good length generalization is an important and difficult problem. The results in this paper can motivate future work on achieving length generalization on more complex tasks, by adopting appropriate data filtering within a self-improvement framework.

**Theoretical Claims:**

There is no theoretical claim.

---

> ### Author Rebuttal · Authors · 2025-04-01
>
> Thank you for your careful review and for recognizing the thoroughness of our experimental design.
>
> **W1 Limitations to simple tasks**
> > This paper is very much limited by the simplicity of the task[...] length of the input corresponds nicely to the difficulty of the problem. However, in many (and more realistic) cases, such a correspondence is not that straightforward.
>
> We agree that for most of the tasks we considered, longer length implies higher difficulty. But for tasks like maze solving, a larger number of nodes do not correspond strictly to harder instances; it is possible to have a smaller maze that is harder due to a higher branching factor. It would indeed be interesting to consider tasks whose difficulty scales constantly or inversely with length. Moreover, we absolutely agree that estimating the difficulty of real-world tasks is an important future direction, and we discuss this point more in the Limitations section of our paper.
>
> For more on real-world applicability, please refer to our general response below.
>
> **W2 Data filtering**
> > Without the data filtering procedure, the self-improvement framework will lead to much worse result. It is unclear how to design good data filtering procedure on more complex tasks.
>
> Indeed data filtering is crucial to the success of our self-improvement framework. One key finding from our experiments is the effectiveness of simple majority vote filtering. This strategy is notable because it does not rely on task-specific heuristics, making it potentially applicable to more complex scenarios.
> Moreover, filtering based on majority voting to ensure self-consistency is a widely-used approach, as we mention in our paper (L265).
>
> **W3 Implications for Complex Settings**
> > Therefore, the implications of the findings presented in this paper to more complex settings is not clear. Besides stating this as a limitation, the authors should at least discuss how the results on simple tasks would be useful for more complex tasks.
>
> We acknowledge your valid concern regarding the implications of our findings for more complex, real-world settings. As we discuss explicitly in our Limitations Section, defining and quantifying task difficulty remains an open and significant challenge in practical applications. However, our results provide valuable insights into systematically scaling task difficulty, an essential step for enabling transformers to tackle more complex real-world problems effectively.
>
> Moreover, our experiments show that models exhibit robustness to imperfect difficulty scheduling, especially when starting from pretrained models. This robustness improves further with increased training rounds. Additionally, we find that decomposing complex problems into intermediate steps enhances the model's ability to generalize. Notably, pretrained models are particularly effective at leveraging these decompositions, highlighting a promising avenue for applying our method to complex tasks.
>
> ---
> ## General response to overlapping concerns:
> > Real-world Applicability and Generalization Beyond Synthetic Tasks.
>
> We acknowledge this valid concern. We selected synthetic tasks for their well-defined difficulty and clear evaluation metrics, which allow for controlled and rigorous analysis.
>
> Additionally, the scope of our chosen tasks aligns with common practice in the length generalization literature. Many influential works (Li et al., 2023;Ruoss et al., 2023;Zhou et al., 2023;McLeish et al., 2024;Kazemnejad et al., 2024;Sabbaghi et al., 2024;Cho et al., 2024;Zhou et al., 2024;Fan et al., 2024) have focused on the problem of generalizing beyond lengths seen during training using synthetic tasks, gaining recognition for providing critical insights into model limitations and capabilities. In fact, this line of research dates back to early works such as Neural GPUs(Kaiser and Sutskever, 2016) and Universal Transformers (Dehghani et al., 2019). We cite these studies to legitimize our methodology, demonstrating that insights derived from synthetic tasks are widely considered meaningful and impactful.
>
> Compared to many prior works, our work considers a broader range of synthetic tasks with varying difficulty, encompassing arithmetic operations, string manipulations, and maze-solving problems. These tasks allow us to stress-test self-improvement in diverse settings and build insights applicable to more complex domains. Although out of the scope of this paper, we plan to extend self-improvement to long-context natural language reasoning benchmarks.
>
> Overall, we believe our paper presents an important step forward. We demonstrate improvements to the existing length generalization literature by providing a method that can continue the trend of generalization indefinitely, on a wider range of tasks. We supplement the work with robust empirical analyses. Our work establishes foundational insights on how self-generated data and iterative self-improvement can extend the capabilities of transformer models.

---

### Official Review · Reviewer_J7z1 · 2025-03-13

**Overall Recommendation:** 3

**Summary:**

This paper proposes and validates a simple and intuitive idea to train a model to solve hard problems that require long reasoning processes. The authors train the models on a task with progressively increasing complexity, leveraging the models' capability to generalize to slightly harder ones for self-improvement. The authors validate the success of this method with a number of synthetic tasks, including a maze puzzle. Analyses highlight the importance of filtering correct self-synthetic data through length filtering and majority voting in this method.

**Claims And Evidence:**

The main claim of this paper is that language models can generalize easy-to-hard and we can leverage this capability to self-improve models to solve progressively complex problems of a task. This claim is supported by convincing evidence of several synthetic tasks including difficult maze-solving.

The author emphasizes that reliable filtering is central to consistent self-improvement and that error avalanche caused by label noise as a key failure case of the self-improvement process. The results Fig. 6 and Fig. 9 justify this.

The rate of self-improvement can be exponential and pretrained models can achieve faster acceleration in easy-to-hard generalization. The evidence in Fig. 12 supports this.

**Essential References Not Discussed:**

Not found.

**Experimental Designs Or Analyses:**

The experiment designs are presented clearly. I am particularly interested in how much difficulty increased in each round of iteration for each task. As mentioned in the Introduction, controlling the weak-to-strong curriculum is crucial, as models require a structured difficulty schedule to avoid catastrophic failure. But the authors only discuss this by directly putting the final setups without explaining the rationales in the Appendix.

**Methods And Evaluation Criteria:**

This paper presents the effectiveness of the methods under synthetic tasks. These tasks make the task difficulties easy to control which might cause concerns about adopting the method to real-world tasks. Besides, it is unclear in what situations the proposed unsupervised filtering method works or fails.

**Other Comments Or Suggestions:**

See above.

**Other Strengths And Weaknesses:**

Strengths:
1. This paper is clearly organized and the experiments are extensive.
2. The claim in this paper is well justified through well-controlled experiments.
3. It is interesting to see unsupervised filtering technique is enough to drive fil

Weaknesses:
1. The method in this paper is limited to synthetic settings with several implicit assumptions: (1) single task with fine-grained controllable difficulties; (2) the existence of (unsupervised) filtering rules that enable near-perfect filtering, without which the accuracy drops obviously after few iterations.

It would be better if the authors can discuss more on how their findings can potentially help facilitate real-world tasks.

2. It is not surprising to see reliable filtering is critical for the success of .

**Questions For Authors:**

See above.

**Relation To Broader Scientific Literature:**

Results in this paper seem to relate to the inference length scaling of reasoning models[1].

It would be interesting to discuss the connection between length scaling during RL and the findings in this paper.

[1] DeepSeek-R1: Incentivizing Reasoning Capability in LLMs via Reinforcement Learning

**Theoretical Claims:**

N/A

---

> ### Author Rebuttal · Authors · 2025-04-01
>
> We appreciate your recognition of our extensive experimental validation and clear presentation. We respond to your questions inline below.
>
> **W1 Limitations to Synthetic Setting**
> > It would be better if the authors can discuss more on how their findings can potentially help facilitate real-world tasks.
>
> Please refer to our general response below in our response to Reviewer MReD (General response to overlapping concerns).
>
> **W2 Filtering**
> > The method in this paper is limited to synthetic settings with several implicit assumptions: ... (2) the existence of (unsupervised) filtering rules that enable near-perfect filtering, without which the accuracy drops obviously after few iterations.
>
> > It is not surprising to see reliable filtering is critical for the success of [.] (note: this comment appears to be cut off)
>
> We agree that filtering is central to the framework. One interesting result is that simple majority voting over models trained with different seeds is surprisingly effective. This unsupervised, task-agnostic approach allows for easy application to a variety of tasks, including more complex ones.
>
> **Difficulty schedule**
> > The method in this paper is limited to synthetic settings with several implicit assumptions: (1) single task with fine-grained controllable difficulties ...
>
> > I am particularly interested in how much difficulty increased in each round of iteration for each task.  As mentioned in the Introduction, controlling the weak-to-strong curriculum is crucial, as models require a structured difficulty schedule to avoid catastrophic failure. But the authors only discuss this by directly putting the final setups without explaining the rationales in the Appendix.
>
> Thank you for pointing out the need to clarify how we selected difficulty increments. Our primary consideration was to ensure data quality sufficient for reliable training in subsequent rounds, thus preventing error avalanches (Section 8). We recognize that our rationale behind difficulty increments could be clearer, and we will explicitly connect the choice of difficulty scaling with the avoidance of error accumulation. Additionally, we will include an experiment demonstrating the consequences of overly aggressive difficulty scaling, further clarifying the importance of difficulty schedule.
> Our existing "accelerated self-improvement" experiments (Section 7.2, Section B.4, Figures 12 & 23) also offer insights into the flexibility of our difficulty schedules, demonstrating the allowable range of difficulty increases with self-improvement rounds.
>
> **Connection to Length Scaling of Reasoning Models**
> > Results in this paper seem to relate to the inference length scaling of reasoning models[1].It would be interesting to discuss the connection between length scaling during RL and the findings in this paper. ([1] DeepSeek-R1: Incentivizing Reasoning Capability in LLMs via Reinforcement Learning)
>
> Thank you for pointing out the potential connection between our results and the inference length scaling observed in reasoning models like DeepSeek-R1. Indeed, both lines of work emphasize iterative and incremental learning paradigms. We add a short discussion of this  connection and its implications in our revised manuscript.

---

### Official Review · Reviewer_vrfN · 2025-03-16

**Overall Recommendation:** 4

**Summary:**

The paper introduces a self-improvement framework for transformer models that enables them to progressively tackle problems beyond the training distribution. Rather than modifying the underlying transformer architecture, the authors leverage an iterative self-training procedure in which a model generates its own training data and incrementally learns from increasingly difficult examples. The work is demonstrated across a range of algorithmic tasks—including arithmetic (both reverse and forward addition, as well as multiplication), string manipulation (copying and reversing), and maze solving. Two key unsupervised data filtering techniques are proposed: length filtering, which removes outputs shorter than an expected threshold, and majority voting, which retains only those examples that a set of independently trained models agree on. The experiments show that models trained with this framework can, for instance, generalize from 10-digit to 100-digit arithmetic problems and similarly extend their capabilities in other tasks.

**Claims And Evidence:**

The paper’s core claims are:

- Self-improvement enables transformers to overcome length and difficulty generalization challenges. They back this up with extensive experimental results on arithmetic (up to 100-digit addition, up to 10-by-10 multiplication), string tasks (copying, reversing up to 120 tokens), and maze pathfinding (from up to 9 hops to 30 hops, or from 30 nodes to 100 nodes). The evidence is convincing: they show consistent gains and near-perfect accuracies over multiple self-improvement rounds.
- Data filtering is critical for stable self-improvement. The authors highlight “error avalanches,” illustrating how low-quality self-labeled data can accumulate and degrade performance. They demonstrate that straightforward filters—length-based filtering and majority voting across multiple models—significantly reduce noise. They also document label noise injection experiments that lead to self-improvement collapse, underscoring the importance of high-fidelity synthetic labels. This is well-supported by thorough ablations.
- Pretrained models accelerate and improve self-improvement. The authors show that starting from larger LLaMA-based checkpoints (1B and 3B parameters) makes it easier to expand to out-of-distribution difficulties. These results, while not deeply analyzed in terms of underlying reasons, are plausible and are backed by side-by-side comparisons of smaller vs. larger models.
- Long-horizon iterative training can lead to exponential improvements. They present an “accelerated self-improvement” schedule where the new round’s difficulty is not just “one step harder” but rather includes all newly mastered tasks. This quickly expands the range of tasks the model can solve. The experimental plots do show faster improvement under these scheduling choices.

**Essential References Not Discussed:**

The paper is well-cited

**Experimental Designs Or Analyses:**

The experimental setup is sound:

- Controlled Difficulty Increase: For each task, a clear definition of difficulty (e.g., number of digits or hops) is provided. The incremental increase per round is systematic.
Ablation Studies: The paper examines the impact of filtering techniques, accelerated curricula, and the role of pretraining.
- Error Avalanche Analysis: By injecting synthetic errors and tracking their impact, the authors provide a useful analysis of how self-improvement can collapse if noise is not managed.

One potential weakness is that all experiments are conducted on synthetic or algorithmic tasks. While these tasks are standard in the literature on length generalization, additional evaluations on more diverse real-world tasks might help demonstrate broader applicability.

**Methods And Evaluation Criteria:**

- Self-Improvement Procedure: The iterative framework alternates between data generation (using the current model) and training with an expanded dataset, where new examples are filtered by length and consensus.
- Data Filtering Techniques: Two unsupervised filters (length filtering and majority voting) are presented to prune noisy outputs.
- Evaluation: Performance is measured by exact-match accuracy using greedy decoding. For arithmetic tasks, generalization is quantified by the maximum digit length (or operand size) that achieves near-perfect accuracy.

**Other Comments Or Suggestions:**

N/A.

**Other Strengths And Weaknesses:**

Strengths:

- The iterative framework is a compelling strategy to push the boundaries of what transformers can generalize.
- By relying on standard transformer architectures without task-specific modifications, the method is both simple and potentially widely applicable.
- The paper presents extensive ablations and analyses, particularly on error propagation and the impact of filtering.

Weaknesses:

- The experiments focus solely on synthetic tasks. More real-world applications would be valuable to assess practical utility. I would expect the authors try their method to adapt a model trained on shorter text sequences to process longer inputs.
- The method’s performance on tasks with more complex or less structured data remains to be explored.
- Why do we need self-improvement while we can use positional encoding and not requires training to generalize to longer sequence? The authors did mention that "While effective in controlled setups, these approaches are often incompatible with large language models (LLMs) in practice, as they introduce task-specific modifications that are difficult to scale across diverse applications." but RoPE, for example is applied on Deepseek and it is not "task-specific".

Overall, this paper presents a solid contribution to understanding how self-generated data and controlled curricula can extend transformer capabilities beyond their initial training distribution. While certain aspects—such as scalability to non-synthetic tasks and formal theoretical analysis—remain open, the empirical results and methodological innovations make this work a significant and promising step forward.

**Questions For Authors:**

- How do you envision scaling this self-improvement framework to tasks beyond synthetic benchmarks? Could you provide examples of potential real-world applications or domains where you expect similar performance gains?
- Have you conducted experiments to assess how sensitive the method is to the choice of filtering thresholds in both length filtering and majority voting? What guidelines can you offer for tuning these parameters in practice?

**Relation To Broader Scientific Literature:**

The paper is well-situated within the current literature:

- It builds on prior work in length generalization (e.g., Anil et al., 2022; Zhou et al., 2023) and transformer modifications.
The self-improvement framework is related to recent studies on self-training and self-refinement in large language models (e.g., Huang et al., 2022; Singh et al., 2023).
- The idea of easy-to-hard curriculum learning connects with literature on curriculum learning and weak-to-strong generalization.
- The work distinguishes itself by showing that without modifying the transformer architecture, one can achieve significant extrapolation capabilities using a controlled self-improvement loop.

**Theoretical Claims:**

The paper is primarily experimental, focusing on empirical performance improvements rather than deep theoretical guarantees.

---

> ### Author Rebuttal · Authors · 2025-04-01
>
> We thank the reviewer for their detailed feedback and for acknowledging that the iterative framework is a compelling strategy to push the boundaries of the generalization capabilities of Transformer models, and recognizing our extensive ablations and analyses. Please find our responses to each of the concerns you have raised below.
>
> **W1,W2, Q1 Real world applications**
> > W1 The experiments focus solely on synthetic tasks. More real-world applications would be valuable to assess practical utility. I would expect the authors try their method to adapt a model trained on shorter text sequences to process longer inputs
>
> > W2 The method's performance on tasks with more complex or less structured data remains to be explored.
>
> > Q1 How do you envision scaling this self-improvement framework to tasks beyond synthetic benchmarks? Could you provide examples of potential real-world applications or domains where you expect similar performance gains?
>
> Please refer to our General Response below in our response to Reviewer MReD (General response to overlapping concerns) regarding the limitations on synthetic tasks.
>
> **W3-1 Necessity of self-improvement vs. positional encodings**
> > Why do we need self-improvement while we can use positional encoding and not requires training to generalize to longer sequence?
>
> While approaches like RoPE indeed improve length generalization, they are complementary rather than replacements. Our self-improvement method is architecture-agnostic, requires no positional encoding modification, and naturally scales across diverse applications without architecture changes. This generality positions our method advantageously, especially when considering broad applicability to existing large pretrained models, avoiding retraining costs or architectural modifications. For example, for our pretrained model experiments, we employed pretrained LLaMA models utilizing RoPE. Hence, the choice of positional encoding is orthogonal to our core contribution. We will clarify and elaborate this point.
>
> **W3-2. On RoPE scaling for long-context**
> > The authors did mention that 'While effective in controlled setups, these approaches are often incompatible with large language models (LLMs) in practice, as they introduce task-specific modifications that are difficult to scale across diverse applications.' but RoPE, for example is applied on Deepseek and it is not 'task-specific'."
>
> RoPE itself is only a positional encoding choice, but we agree that RoPE scaling, used in DeepSeek, is proposed as a way to generalize to longer contexts without retraining the model. However, existing works on long context model evaluation [https://openreview.net/pdf?id=293V3bJbmE] shows that RoPE still struggles to achieve reliable performance on OOD input lengths. Furthermore, applying RoPE scaling on our synthetic tasks did not help with length generalization in our experiments. We will add our result on using RoPE scaling in the Appendix.
>
>
> **Q2. Sensitivity of Filtering Thresholds**
> > Have you conducted experiments to assess how sensitive the method is to the choice of filtering thresholds in both length filtering and majority voting? What guidelines can you offer for tuning these parameters in practice?
>
> We agree this is important and will add further clarification in the paper. Based on our current observations, the impact of length filtering is task-dependent. In tasks like multiplication where errors tend to drop intermediate steps, the threshold is less sensitive. In contrast, tasks like forward addition(where incorrect outputs are typically just 1–2 digits short), stricter filtering is needed to avoid error propagation. Regarding majority voting, we plan to expand our ablations to show how the number of models affects filtering effectiveness. As indicated in prior work (e.g., https://arxiv.org/pdf/2306.15400), even a small number of highly accurate training examples can suffice for strong length generalization. Therefore, filtering for high-precision data by strong consensus among multiple models - even at the cost of sample quantity - can be beneficial. We thank the reviewer for suggesting this analysis and will include it in our revision.

---

### Decision · Program_Chairs · 2025-05-01

**Decision:**

Accept (poster)

**Comment:**

**Summary of the Paper:**
This paper introduces a self-improvement framework for transformer models, enabling them to progressively tackle problems beyond their training distribution. The approach leverages iterative self-training, where models generate their own training data and learn from increasingly difficult examples. Demonstrated across tasks like arithmetic, string manipulation, and maze solving, the method shows significant improvements in out-of-distribution performance, such as generalizing from 10-digit to 100-digit addition.

**Strengths and Weaknesses Highlighted by Reviewers:**
- **Strengths:** Reviewers praised the innovative self-improvement framework and its solid empirical results across diverse tasks. The method's simplicity and potential broad applicability were highlighted, along with thorough ablations and analyses on error propagation and filtering techniques.
- **Weaknesses:** Concerns were raised about the focus on synthetic tasks and the practical relevance of the findings. Some reviewers questioned the scalability of the method to real-world applications and the computational cost involved.

**Rebuttal and Discussion with Reviewers:**
The authors provided detailed responses, emphasizing the foundational insights gained from synthetic tasks and the potential for future extensions to real-world scenarios. They clarified the effectiveness of their filtering techniques and addressed concerns about computational costs and the choice of positional encoding. Reviewers acknowledged the thorough rebuttal, with most maintaining their positive evaluations, while one reviewer (MReD) remained skeptical about practical implications.

**Recommendation for Acceptance:**
Given mostly positive (although not unanimous) assessments, we recommend accepting this paper as it presents a step forward in understanding and improving transformer models' generalization capabilities. While reviewer MReD raised valid concerns about practical relevance, narrow results in synthetic settings do have a place in science, as they are often crucial initial steps towards broader understanding and future applications. Thus, we believe the paper's innovative approach, extensive empirical validation, and detailed analyses make it a valuable contribution to the field.